RESEARCH

# Comparative genomics identifies thousands of candidate structured RNAs in human microbiomes

Brayon J. Fremin[1] and Ami S. Bhatt[1,2]*

* Correspondence: asbhatt@stanford.edu
[1]Department of Genetics, Stanford University, Stanford, CA 94305, USA
[2]Department of Medicine (Hematology), Stanford University, Stanford, CA 94305, USA

## Abstract

**Background:** Structured RNAs play varied bioregulatory roles within microbes. To date, hundreds of candidate structured RNAs have been predicted using informatic approaches that search for motif structures in genomic sequence data. The human microbiome contains thousands of species and strains of microbes. Yet, much of the metagenomic data from the human microbiome remains unmined for structured RNA motifs primarily due to computational limitations.

**Results:** We sought to apply a large-scale, comparative genomics approach to these organisms to identify candidate structured RNAs. With a carefully constructed, though computationally intensive automated analysis, we identify 3161 conserved candidate structured RNAs in intergenic regions, as well as 2022 additional candidate structured RNAs that may overlap coding regions. We validate the RNA expression of 177 of these candidate structures by analyzing small fragment RNA-seq data from four human fecal samples.

**Conclusions:** This approach identifies a wide variety of candidate structured RNAs, including tmRNAs, antitoxins, and likely ribosome protein leaders, from a wide variety of taxa. Overall, our pipeline enables conservative predictions of thousands of novel candidate structured RNAs from human microbiomes.

## Background

Microbial structured RNAs are involved in biological processes ranging from *cis*-antisense regulation to targeted regulation of diverse sets of genes [1]. We define structured RNAs as any non-coding RNA with a conserved secondary structure across taxa [2]. Given the exciting demonstrated roles that structured RNAs play in bioregulation, there is enthusiasm for developing methods to more comprehensively identify structured RNAs. Previously, comparative genomics was applied for this purpose to high effect, revealing hundreds of novel candidate structured RNA motifs [2]. In general, these comparative genomic approaches involve extensive clustering of subsequences of intergenic regions from many organisms followed by structural motif prediction [3, 4]. These motifs are then scored based on observance of nucleotide covariation; this

covariation suggests that when one nucleotide of a predicted structured motif undergoes mutation, a compensatory mutation of the complementary base-pairing nucleotide is also observed, preserving the overall structure [3, 4]. In the last decade, hundreds of structured RNAs in bacteria have been predicted using comparative genomics [2, 5]. The small subset of these RNAs that have been validated and carefully characterized display an exciting array of activities, ranging from functioning as *trans*-acting ncRNAs to self-cleaving ribozymes to riboswitches [6, 7].

While powerful, a drawback of this comparative genomics approach is that in any given experiment only a select set of intergenic regions are typically considered; this is done to accommodate computational limitations and reduce the false positive rate. For example, comparative genomics approaches often subset intergenic regions to bias towards known classes of structured RNAs, like ribozymes or specific types of riboswitches, which tend to be found in specific genomic contexts [2]. As a consequence of computational limitations and subsetting of intergenic regions, much of the intergenic search space in microbes remains unexplored. Thus, carefully investigating these regions presents an opportunity to discover new structured RNAs; however, this must be approached with caution as larger-scale studies can potentially introduce more false positives and make prediction of rare RNAs difficult [2].

Though computationally expensive, mining intergenic regions of large metagenomic datasets is necessary for us to find structured RNAs in these largely unexplored regions of microbial genomes. One of the rate limiting steps in a pipeline to predict candidate structured RNAs is performing all-versus-all BLASTn against all intergenic regions of interest [2, 5]. Developments such as high speed BLASTN (HS-BLASTN) [8] can make such a rate limiting step more computationally feasible (in terms of CPU-hrs) to perform at large-scale. In this work, we perform an all-versus-all HS-BLASTN [8] on hundreds of millions of intergenic regions predicted from Human Microbiome Project phase 2 (HMP2) [9] metagenomic data, resulting in billions of significant pairwise homologies between different intergenic regions. We cluster these homologous regions and predict millions of possible structured motifs. Using conservative scoring methods, we propose thousands of candidate structured RNAs and validate expression of hundreds of them.

## Results

### Workflow of high-throughput predictions of candidate structured RNAs

To identify candidate structured RNAs, we created the following workflow (Fig. 1a). First, we predicted 214,794,089 intergenic regions (~ 20 million base pairs) from HMP2 [9] by identifying coding regions using Prodigal [10] and considering those regions not predicted to be coding. To determine which regions shared homology with each other, we performed all-versus-all HS-BLASTN [8] on all intergenic regions ($E$ value 0.05). We filtered out conserved regions that were short (< 30 base pairs) or too similar to each other (BLAST bit-score > 100 or percent identity = 100) [5]; the latter filter was used as closely related sequences are unlikely to provide valuable covariation information. Many of these thresholds were inspired by Weinburg et al. [5]. From the resulting 6,241,850,878 pairwise homologies from HS-BLASTN [8], we clustered homologous regions using overcluster2 [2] with default settings, yielding 7,878,825 clusters. We structurally aligned these regions using CMfinder (version 0.4.1) [17].

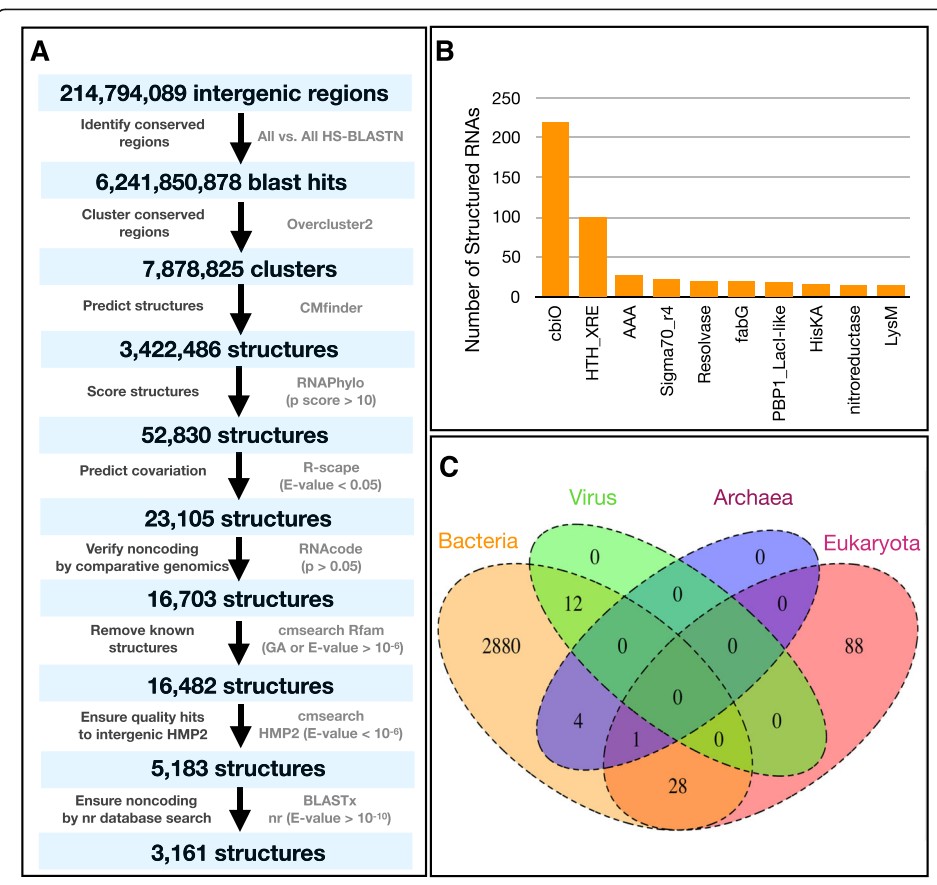

**Fig. 1** Prediction and characterization of candidate structured RNAs. **a** Workflow of candidate structured RNA prediction. Genes in HMP2 [9] are annotated with Prodigal [10]. Homologous intergenic regions are identified using HS-BLASTN [8]. Conserved regions are clustered and scored using RNAphylo [2], R-scape [11], and RNAcode [12]. These putative structured RNAs are filtered to exclude those already found in Rfam 14.3 [13]. Candidate structured RNAs are searched against HMP2 using cmsearch [14] to ensure strong, unique hits were observed for each proposed structure. To triple check candidate structured RNAs are in non-coding regions, we search regions against the nr database using BLASTx [15]. This results in 3161 candidate structured RNAs, in addition to 2022 candidate structured RNAs with regions overlapping the nr database. **b** Histogram displaying the ten protein domains that are most commonly found in the genomic neighborhoods (within 5 kilobases) of the 3161 candidate structured RNAs. **c** Venn diagram displaying the taxonomic distributions of the 3161 candidate structured RNAs at the domain level based on One Codex [16] analysis of contigs on which they are found

To assess the quality of these structured RNA predictions, we performed two scoring methods. First, we performed RNAPhylo [2], which uses a phylogenetic model to assign probabilistic scores to alignments, narrowing the alignment files down to 52,830 motifs (RNAPhylo p score > 10). Second, we assessed which structured motifs exhibited significant covariation in nucleotides ($E < 0.05$) using R-scape [11], resulting in 23,105 motifs with significant covariation. To further determine if any of these potential structured RNAs could encode a protein, we performed RNAcode [12], a comparative genomics approach to predict coding regions using multiple sequence alignments. We found that 16,703 possible structured RNAs were unlikely to be coding (RNAcode p value < 0.05). We performed cmsearch from Infernal [14] across intergenic regions against Rfam 14.3 [13] and found that 221 of the possible structured RNAs are previously known (GA cutoff or $E$ value $< 1 \times 10^{-6}$). Candidate structured RNAs overlapping

with any region with a known structure in Rfam [13] were discarded, finally yielding 16,482 putative models of structured RNAs (Additional file 1: Table S1). To ensure these models were of high quality when used as search queries, we performed cmsearch [14] for these 16,482 possible structured RNAs against HMP2 [9] intergenic regions. We aimed to avoid proposing the same models multiple times or proposing structures not yielding significant cmsearch [14] results. For example, it is possible that two clusters formed by overcluster2 in our pipeline, though distinct based on HS-BLASTN [8] thresholds, are similar enough sequences to yield roughly the same structure and hit the same regions. We ultimately proposed 5183 possible structured RNAs that uniquely and significantly hit intergenic regions (cmsearch $E$ value $< 1 \times 10^{-6}$).

As a final measure to ensure these predictions are in non-coding regions, we performed BLASTx [15] on all of these regions against the nr database (non-redundant reference protein sequences). This resulted in 3161 candidate structured motifs (BLASTx $E$ value $< 1 \times 10^{-10}$) that were not found in regions predicted to be coding per the nr database. To calculate the false discovery rate (FDR) of these predictions, we first shuffled these alignments using SISSIz [18]. We subjected these shuffled alignments to the complete pipeline including CMfinder [17], RNAphylo [2] (p-score $> 10$), and R-scape [11] ($E < 0.05$) to determine the likelihood that these regions might satisfy our thresholds for being predicted as a candidate structured RNA. Only 89 of these 3161 shuffled alignments were predicted to be structured RNAs, suggesting an FDR of 0.028. Interestingly, the 2022 possible structured RNAs with significant BLASTx [15] hits to the nr database also had a low FDR rate of 0.056 (113/2022). These calculations of FDR are likely an underestimation or lower bound of the true FDR. Finally, we sought to address the possibility that candidate structured RNAs may overlap with possible small proteins, which are often unannotated by standard annotation pipelines. To investigate this, we applied SmORFinder [19], a machine learning approach to predict small proteins based on characteristics of previously predicted small proteins, and found that 99 of the candidate structured RNAs may overlap small genes (Additional file 2: Table S2).

Next, we characterized the 3161 candidate structured RNAs and the 2022 candidate structures that overlapped with the nr database by their length, taxonomy, and genomic neighborhood (Additional file 1: Table S1). The 3161 candidate structured RNAs ranged from 27 to 219 base pairs in length (Additional file 1: Table S1). To determine if certain genes were more often present in the vicinity of structured RNAs, we performed a genomic neighborhood analysis. Specifically, we determined which genes were present within 5 kilobases of every candidate structured RNA; we found that the protein domains cbiO and HTH-XRE were the most frequently annotated genes found within close proximity to our novel structured RNAs (Fig. 1b), with 219 and 100 respective candidate structured RNAs found in these genomic contexts. Genomic context, while occasionally useful, does not suggest function of candidate structured RNAs. While these 3161 candidate structured RNAs were predominantly found in bacteria, some were identified in other domains: 12 structures were found in viruses, 5 in archaea, and 117 in eukaryota (Fig. 1c). Notably, the number of structures classified to viruses is likely underestimated as many structures encoded in phage genomes may classify to their host bacteria. The genomic neighborhood analysis can be a useful way to determine if structured RNAs may be found on phage as these structures are often surrounded by phage-specific protein domains. For example, there are 28 candidate

structured RNAs most commonly found near protein domains with "integrase," "sortase," or "phage" in their names (Additional file 1: Table S1). Though we are showing all classified structures (Fig. 1c), there is variability in accuracy of classifications; this is reflected in the percentage of mapped k-mers from One Codex [16] (Additional file 1: Table S1). Finally, we carried out a similar analysis on the 2022 structures that overlapped the nr database and we report these results as a separate, supplemental resource (Additional file 1: Table S1).

## Comparative genomics reveals a diverse collection of candidate structured RNAs

Due to the large size of this collection, it is unfortunately not feasible to discuss every candidate structured RNA in great detail. Thousands of interesting structures can be found in our collection (Additional file 1: Table S1) and can be filtered to focus on specific organisms and genomic contexts of interest. To showcase diversity within the 3161 candidate structured RNA, we highlight a few examples from this resource. For example, HMP2_2419 is a 126 base pair (bp) candidate structured RNA (Fig. 2a) that

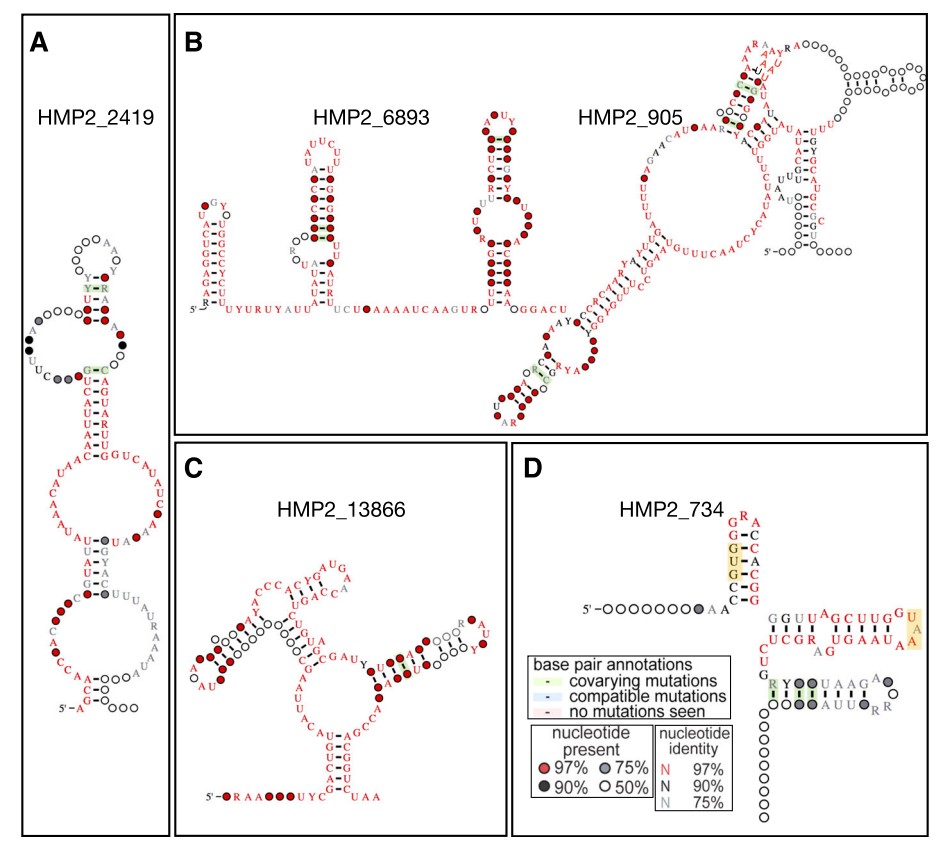

**Fig. 2** Diversity of candidate structured RNAs. **a** Structure diagram, showing the consensus features of HMP2_2419, a candidate structured RNA that associates with tRNA-ligases. **b** Structure diagrams of HMP2_6893 and HMP2_905, candidate structured RNAs that overlap in opposing orientations that are in the genomic neighborhood of *uvrC*. **c** Structure diagram of HMP2_13866, a candidate structured RNA found in *Malassezia*. **d** Structured diagram of HMP2_734, a candidate structured RNA that is likely a novel tmRNA. The legend in **d** applies to all structure diagrams in this manuscript to indicate nucleotide presence, nucleotide identity, and covariation. Yellow shading indicates the start and stop site of the small gene within the tmRNA

was taxonomically difficult to classify as only 1.5% of k-mers on average could be classified from the contigs it was found on using One Codex [16]. HMP2_2419 was most commonly found near (within 5 kilobases) of Valine tRNA-ligase (*valS*) in 290 instances, Leucine tRNA-ligase (*leuS*) in 146 instances, and Isoleucine tRNA-ligase (*ileS*) in 143 instances (Additional file 1: Table S1). Although structurally distinct, seven other candidate structured RNAs were found near *valS* and other tRNA-ligases, including HMP2_2187, HMP2_2353, HMP2_2391, HMP2_3524, HMP2_6788, HMP2_8154, and HMP2_9059 (Additional file 1: Table S1).

HMP2_6893 (139 bp) and HMP2_905 (205 bp) are candidate structured RNAs that overlap each other in opposing strand orientations (Fig. 2b). In fact, 301 candidate structured RNAs overlapped one or more other candidate structured RNAs in the opposing orientation (Additional file 3: Table S3). This may be a system in which one candidate structured RNA regulates the other, as is seen in the SdsR and RyeA toxin-antitoxin system [20, 21]. HMP2_6893 and HMP2_905 appeared to be exclusive to *Prevotella* in HMP29 and were mostly found near UvrABC endonuclease subunit C (*uvrC*). Four other candidate structured RNAs, HMP2_3411, HMP2_3634, HMP2_598, and HMP2_7697, were also most commonly found near *uvrC* in other organisms, though they are not associated with an overlapping structure given our prediction thresholds (Additional file 1: Table S1).

HMP2_13866 (121 bp) is a candidate structured RNA that appears exclusively in *Malassezia*, a fungal member of the microbiome, in HMP2 [9]. HMP2_13866 was mostly found near NADH dehydrogenase subunit 2 (Fig. 2c, Additional file 1: Table S1). HMP2_734 (84 bp) is a candidate structured RNA that is taxonomically difficult to classify with an average k-mer mapping of 2.29% (Fig. 2d, Additional file 1: Table S1). Interestingly HMP2_734 occurred 36 times in HMP2 [9]; however, in only one of those instances was it predicted to be the tmRNA *ssrA* by Aragorn [22]. Upon inspecting the structure, it indeed appears to be a novel tmRNA with a target peptide code VGTTGLAW*.

### Candidate structured RNAs that are likely *cis*-acting

To identify candidate structured RNAs that may be *cis*-acting, we determined which candidate structured RNAs were found in the potential 5′ UTR of genes. Though we do not know the outermost boundaries of 5′ UTRs, we hypothesized that candidate structured RNAs often found within 25 base pairs of the 5′ end of genes may regulate the downstream gene [2] (Additional file 4: Table S4). The most commonly identified protein domain directly downstream of candidate structured RNAs was HTH_XRE, with 15 candidate structured RNAs found in the potential 5′ UTR of genes with the HTH_XRE domain (Additional file 4: Table S4). Overall, we identified 508 candidate structured RNAs that occur directly upstream (within 25 bp) of genes. For example, HMP2_2105 (86 bp) and HMP2_254 (51 bp) are candidate structured RNAs found in *Treponema* and *Collinsella*, respectively, directly upstream of the 50S ribosomal protein L7/L12 (Fig. 3a). HMP2_39 (49 bp) and HMP2_3043 (57 bp) are candidate structured RNAs found in mostly *Alistipes* and *Treponema*, respectively, directly upstream of the 50S ribosomal protein L20 (Fig. 3b). HMP2_1435 (64 bp) is a candidate structured RNAs found in *Cardiobacterium* directly upstream of the 50S ribosomal protein L10 (Fig. 3c). These 50S ribosomal proteins are known to contain ribosome protein leaders in their 5′ UTRs to

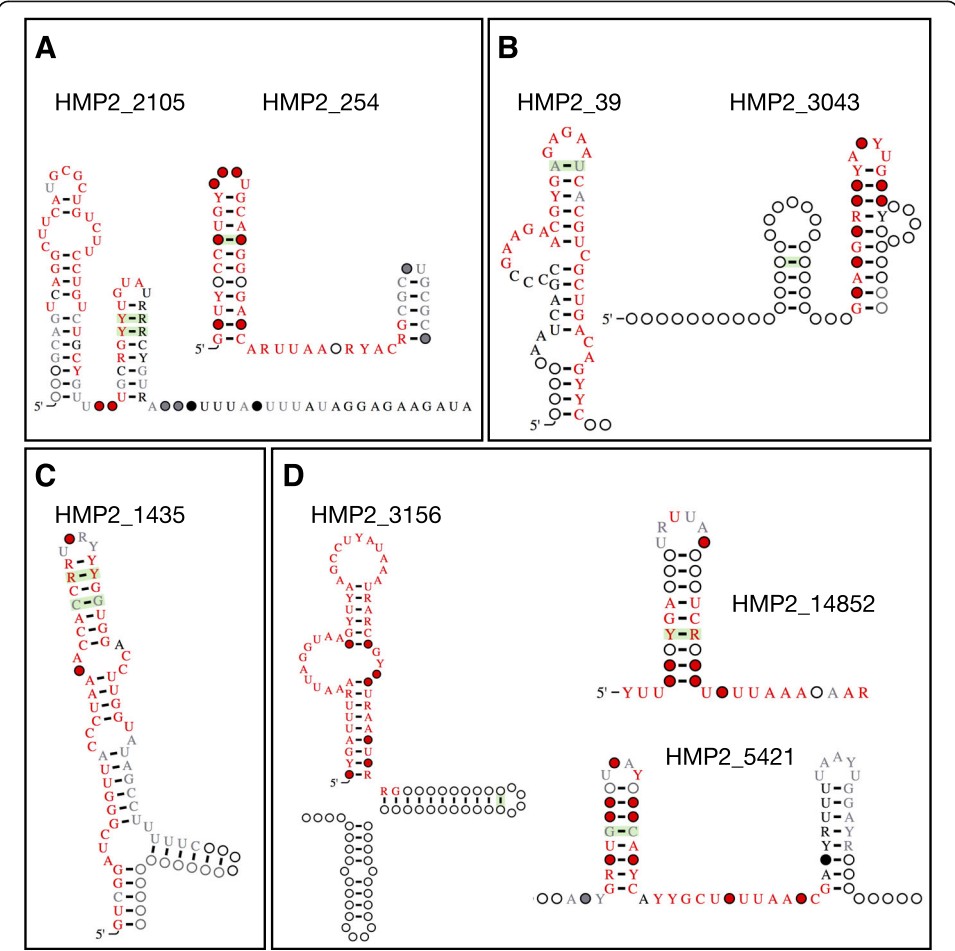

**Fig. 3** Potentially *cis*-regulatory candidate structured RNAs. **a** Structure diagrams of HMP2_2105 and HMP2_254, candidate structured RNAs typically found directly upstream of 50S ribosomal protein L7/L12, possibly ribosome protein leaders. **b** Structure diagrams of HMP2_39 and HMP2_3043, candidate structured RNAs typically found directly upstream of the 50S ribosomal protein L20, possibly ribosome protein leaders. **c** Structure diagram of HMP2_1435, a candidate structured RNAs typically found directly upstream of the 50 S ribosomal protein L10, possibly a ribosome protein leader. **d** Structure diagrams of HMP2_3156, HMP2_5421, and HMP2_14852, candidate structured RNAs often found directly upstream of the *ToxN* toxin, possibly *ToxI* antitoxins

regulate their expression in other bacteria [23, 24], suggesting that these candidate structured RNAs may be ribosome protein leaders. HMP2_3156 (106 bp), HMP2_5421 (68 bp), and HMP2_14852 (38 bp) are candidate structured RNAs found in *Clostridiales* and *Leptotrichia*, respectively, found directly upstream of the *ToxN* toxin (Additional file 4: Table S4, Fig. 3d). The antitoxin, *ToxI*, is a structured RNA often found directly upstream of *ToxN* [25], suggesting that these candidate structured RNAs may be the antitoxin *ToxI*.

### Candidate structured RNAs with palindromic characteristics

We identified candidate RNA structures that are found on both the forward and reverse strand of the same region, suggesting the structures are palindromic. In the finalized set of 3161 candidate structured RNAs, 250 of them were palindromic (Additional file 3: Table S3). There are a few known classes of structured RNAs known to be palindromic, such

as repetitive extragenic palindromic (REP) sequences [26] and Rho-independent terminators [27, 28]. We predicted that 230 of the 250 palindromic candidate structured RNAs could form terminator stem-loop structures using ARNold [27, 28] (Additional file 3: Table S3), which predicts rho-independent terminators in nucleic acid sequences. This suggests that a majority of these palindromic candidates may contain intrinsic terminators. The remaining 20 candidate structured RNAs may be REP sequences or another class of structured RNAs. For example, HMP_4078 (109 bp) is a palindromic candidate structured RNA (Fig. 4a) that was not predicted to contain an intrinsic terminator (Additional file 3: Table S3). It was most commonly found in *Bacteroidetes* and located near bacteriophage related protein domains such as the D5 N terminal domain and phage integrase domain. HMP_4205, HMP2_1065, and HMP2_1032 were all predicted to be Rho-independent terminators and may transcriptionally control the genes they associate with (Fig. 4b–d). HMP2_4205 (132 bp) is a palindromic candidate structured RNA (Fig. 4b) mostly found in *Bacteroidetes* and commonly located near genes that contain tetratricopeptide repeats (TPR). HMP2_1065 (80 bp) is a palindromic candidate structured RNA (Fig. 4c) entirely found in *Bacteroidetes* and most commonly found near the Holliday junction branch migration protein (*RuvA*). HMP2_1032 (106 bp) is a palindromic candidate structured RNA (Fig. 4d) also entirely found in *Bacteroidetes* and most commonly found near periplasmic protein *TonB*.

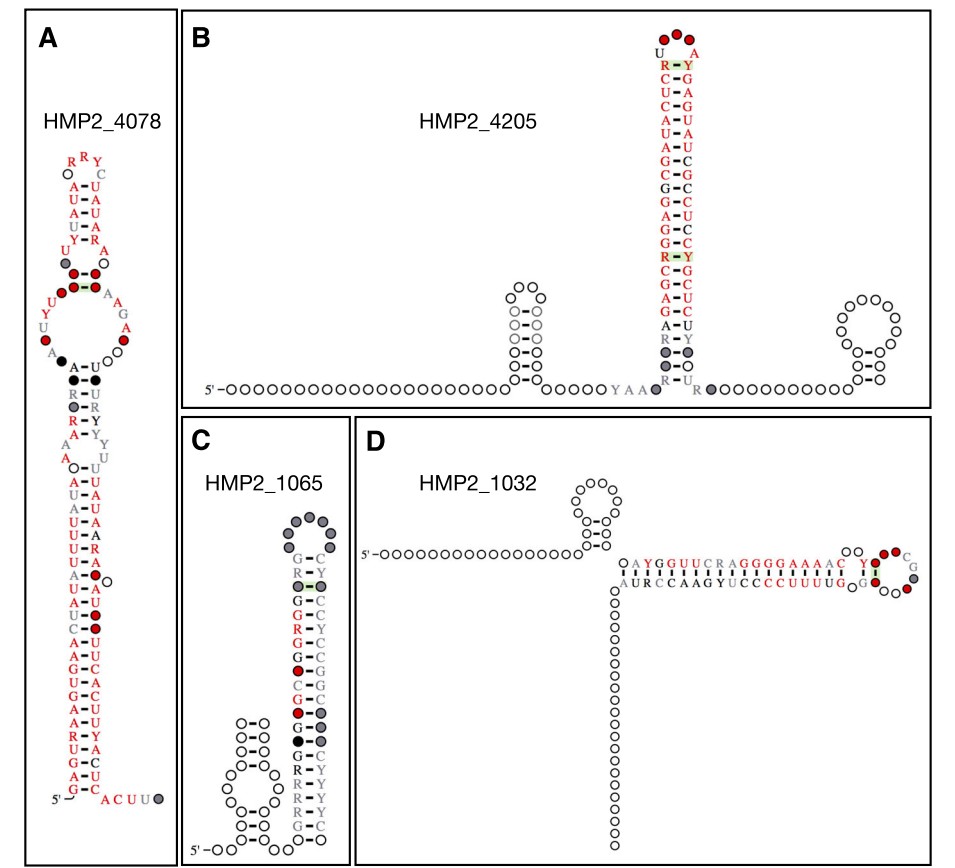

**Fig. 4** Candidate structured RNAs with palindromic characteristics. **a–d** Structure diagrams of HMP2_4078, HMP2_4205, HMP2_1065, and HMP2_1032, respectively, which are candidate structured RNAs that exist on both strands of the same region

### Expressed candidate structured RNAs in fecal microbiomes

To prioritize interesting candidate structured RNAs, we leveraged RNA-Seq, a powerful method to determine which structures are expressed in the fecal microbiome. In this study, we analyzed metagenomic sequencing data and RNA-Seq data without fragment size selection performed on four taxonomically diverse fecal samples from four different human subjects (A—healthy adult, B—patient with hematological disorder undergoing treatment, C—patient with cancer undergoing treatment, D—patient with Alzheimer's disease) [29]. First, metagenomic DNA sequencing data from four human fecal samples was obtained; these data had previously been subjected to computational assembly and the resultant contigs (longer contiguous DNA sequences that are assembled "in silico" from smaller DNA fragments) were collected [29]. RNA-Seq without fragment size selection was previously performed on these same samples and the resultant reads were aligned to the de novo assemblies [29].

First, we searched for the 2022 candidate structured RNAs that contained significant hits to the nr database across these assemblies. We identified 352 of these candidate structured RNAs in the metagenomic assemblies from these samples and found that 59 of them were transcribed at or above an arbitrary threshold of 20 reads per kilobase million (RPKM) (Additional file 5: Table S5). For example, HMP2_8626 (40 bp) is a candidate structured RNA found both in *Firmicutes* and *Bacteroidetes* that overlaps with *pflA*, pyruvate formate-lyase 1-activating enzyme (Fig. 5a). We next searched for expression of the 3161 candidate structured RNAs. We identified 564 of these candidate structured RNAs in the assemblies [29] and calculated that 98 of them were transcribed at RPKM > 20 (Additional file 5: Table S5). For example, HMP2_1881 (108 bp) is a candidate structured RNA predominantly found in *Bacteroidetes* commonly found near radical SAM protein domains. We found HMP2_1881 13 times in our assemblies and it was expressed all 13 times (Additional file 5: Table S5, Fig. 5b). HMP2_7457 (67 bp) is a candidate structured RNA predominantly found in *Firmicutes* and is most often found near the transcriptional repressor domain *PBP1_LacI-like* as well as commonly found near phage domains like integrases (Additional file 1: Table S1, Fig. 5c). HMP2_13009 (56 bp) is a candidate structured RNA found in *Firmicutes* and most often near *Cas1* and *Cas2* protein domains (Additional file 5: Table S5, Fig. 5d). It is a repeating structure and was predicted by minCED (https://github.com/ctSkennerton/minced) to overlap a CRISPR array. This suggests HMP2_13009 is a new model for the structured RNA repeat of a CRISPR/Cas9 system.

### Discussion

Even though structured RNAs in microbes play key roles in bioregulation, many structured RNAs remain undiscovered. It is computationally expensive to build models for new structured RNAs at a large-scale without substantially limiting the search to specific intergenic regions [2]. In this work, we performed computationally intensive, large-scale predictions of candidate structured RNAs across all intergenic regions predicted from HMP2 [9]. Our pipeline analyzed hundreds of millions of intergenic regions, identified millions of clusters of conserved intergenic regions, created millions of candidate motifs, and conservatively scored those motifs with phylogenetic models and covariation statistics to predict thousands of candidate structured RNAs.

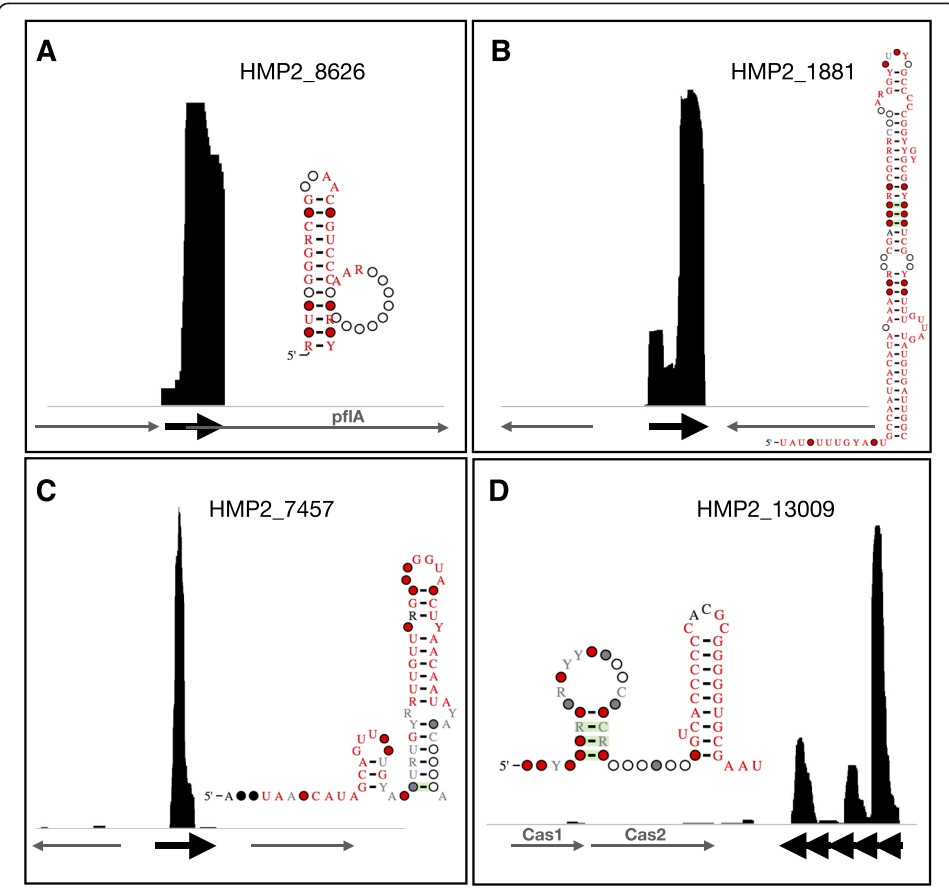

**Fig. 5** Candidate structured RNAs that are expressed in the fecal microbiome. **a** Structure diagram and RNA-Seq IGV plot of HMP2_8626, a candidate structured RNA from the 2022 set that overlaps the nr database, specifically overlapping the gene *pflA*. Black arrows indicate candidate structured RNAs. Gray arrows indicate predicted genes. Unlabeled genes are those predicted to be hypothetical. **b** Structure diagram and RNA-Seq IGV plot of HMP2_1881, a candidate structured RNA from the 3161 set often found near radical SAM protein domains. **c** Structure diagram and RNA-Seq IGV plot of the candidate structured RNA HMP2_7457 often found near *PBP1_LacI-like* and integrase protein domains. **d** Structure diagram and MetaRibo-Seq [30] IGV plot of HMP2_13009, a candidate structured RNA found near *Cas* proteins likely representing the repeats in a CRISPR array

A key tool that enabled us to predict candidate structured RNAs at a large-scale was HS-BLASTN. A key distinction between HS-BLASTN and BLASTn is that HS-BLASTN utilizes a new database-derived lookup table. Unlike BLASTn, it loads the resulting index into memory and thus requires significantly more RAM. To perform all vs. all HS-BLASTN in the workflow required roughly 35,000 CPU-hrs at 90 GB of RAM. We estimated that BLASTn would have required an order of magnitude less RAM but also over an order of magnitude more CPU-hrs. Though this work was still computationally intensive, it would have otherwise been too intensive to realistically perform in our high performance computing environment without HS-BLASTN.

While infeasible to discuss thousands of candidate structured RNAs in detail, we instead highlighted some interesting predictions. We discovered candidate structured RNAs across all taxonomic domains within the human microbiome. We were able to identify many different classes of candidate structured RNAs, including tmRNAs, *cis*-regulatory structured RNAs including putative ribosome protein leaders and antitoxins,

intrinsic terminators, and repeat regions in CRISPR arrays. We further use RNA-Seq from human fecal microbiomes [29] to validate expression and highlight interesting candidate structured RNAs.

While this approach has the advantage of being high throughput, it does have several limitations. First, it can be difficult to distinguish between non-coding and coding regions in microbial genomes. This is evident even in our existing analyses: regions that were not predicted to be coding by Prodigal [10] or RNAcode [12] still contained significant homology to proteins in the nr database, suggesting that the region could, indeed, encode a protein. Of course, significant homology to the nr database also does not necessarily guarantee a region is coding. Additionally, it is also possible that regions that share no homology to the nr database could still be coding. Small genes, for example, are often overlooked and absent in databases. We cannot guarantee if a candidate structure RNA is truly in an intergenic region. Second, this work does not validate RNA structures using methods like FragSeq or SHAPE-Seq [31–33]. We only validate expression in fecal microbiomes of some candidate structured RNAs. Third, the false-negative rate of our work is likely high as we disregard many structures based on our conservative cutoffs. Fourth, the true FDR of these candidate structured RNAs is likely higher than our estimations. Since we predict a diverse set of candidate structured RNAs, we cannot expect shuffling of alignments to adequately control for all biological features. Additionally, we are only shuffling alignments specific to candidated structured RNAs and not shuffling the entire intergenic space. The FDR we estimate in this work is best interpreted as a lower bound of the true FDR. Finally, unlike previous structured RNA discovery approaches, which use comparative genomics along with genomic neighborhood information to select for specific predicted functions such as riboswitches, the candidate structured RNAs we predict in this work can be less straightforward to functionally characterize. For example, the candidate structured RNAs we propose could be *trans*-regulatory and regulate genes found on different contigs in the sample. The genomic neighborhood analyses (5 kb both directions and 25 bp upstream of genes) are biased towards protein domains of higher frequencies in metagenomes and provide no insight on candidate structured RNAs that are *trans*-regulatory. Unless we can make very strong cases for specific classes of candidate structured RNAs, such as the tmRNA and CRISPR repeat examples in this manuscript, most functional assignments would be speculative, at best, and further investigation is certainly warranted.

These limitations notwithstanding, this work provides a resource of 3161 candidate structured RNAs (FDR = 0.028). We provide an additional 2022 candidate structured RNAs that may overlap genes present in the nr database (FDR = 0.056). For all of these candidate structured RNAs, we further characterized them by length, taxonomy, genomic neighborhood, proximity to 5′ of genes, and expression levels, which we anticipate researchers will use to prioritize structures of interest to their lab. Future work will include validating these overlapping candidate structures to determine what roles they play. We proposed 508 structures directly upstream of specific genes, which suggests a role in possible *cis*-regulation, although follow-up work is necessary to determine whether or not *cis*-regulation actually occurs. Being able to truly classify these candidate structures into ribozymes, riboswitches, tRNAs, antitoxins, *trans*-acting ncRNAs, or many other structures will require combined efforts of many labs. We

additionally anticipate researchers will take inspiration from this approach to predict additional candidate structured RNAs employing exciting new additional approaches for clustering candidate RNA structures, such as using GraphClust2 [34], on a large-scale. Overall, we hope that this resource of thousands of structured RNAs enables future functional characterization and mechanistic dissection of a wide range of new structured RNAs from previously unexplored microbial genomic regions.

## Methods

### Data download

All data used in this study are publicly available. Contigs from the 1773 HMP2 metagenomes containing at least 5 Mbp of total contig sequence were downloaded from https://www.hmpdacc.org/hmasm2. The samples used for the RNA-Seq analysis can be found under BioProject accession no. PRJNA510123.

### Comparative genomics workflow

We predicted genes in HMP2 [9] using Prodigal [10]. We inferred which regions were intergenic using bedtools [35]. We performed all-versus-all HS-BLASTN [8] on these intergenic regions using default settings, filtering out regions that were shorter than 30 base pairs, 100% identity to each other, were assigned $E$ values greater than 0.05, or were assigned bit-scores greater than 100. HS-BLASTN [8] requires more RAM as the database size increases. To make HS-BLASTN [8] more computationally feasible, we split the intergenic regions into 100 roughly equally sized smaller fasta files in an unbiased manner and created 100 smaller databases. We ran HS-BLASTN [8], querying all intergenic sequences against all 100 databases. We combined the results from all searches and then clustered similar subsequences based on HS-BLASTN [8] scores using a single-linkage clustering algorithm called overcluster2 (Weinberg, Z., unpublished open-source software, available at http://weinberg-overcluster2.sourceforge.io) using default settings. Clusters were structurally aligned using CMfinder version 0.4.1 [17]. Motifs were scored using RNAPhylo [2] with a p-score cutoff of greater than 10. Motifs were further scored for significant covariation ($E < 0.05$) using R-scape [11] with default settings. When CMfinder [17] proposed multiple motifs for a region that met p-score and covariation cutoffs, the motif with the highest p-score was chosen. To remove conservation that can be explained as coding, RNAcode [12] was applied using default settings and regions with $p$ values greater than 0.05 were retained. We performed cmsearch [14] across the HMP2 [9] intergenic regions against Rfam 14.3 [13] using the GA cutoff or $E$ value greater than $1 \times 10^{-6}$ of a known structure. If any region contained an overlap with Rfam [13] structures and a candidate structured RNA, we discarded the candidate. We similarly performed cmsearch of candidate motifs against HMP2 [9] to ensure they uniquely and significantly ($E$ value $< 1 \times 10^{-6}$) hit regions in HMP2 [9]. We performed BLASTx [15] on all of these regions against the nr database, filtering out those with $E$ values greater than $1^{-10}$. RNA structures were drawn using R2R [36], which was run by default using R-scape [11] previously. These diagrams only highlight covariation that was predicted to be significant by R-scape [11]. Finally, Prokka v1.12 [37] was used on HMP2 [9] using the –meta option, primarily to identify

if candidate structured RNAs may be tRNAs, rRNAs, or CRISPR arrays. Transfer RNAs were predicted by Aragorn [22].

### False discovery rate (FDR) estimates

We shuffled these alignments using SISSlz [18]. To the shuffled alignments, we performed the same pipeline, including CMfinder [17], RNAphylo [2], and R-scape [11] with the same thresholds as above. The number of shuffled alignments that met the thresholds of a candidate structured RNA was divided by the number of alignments considered to calculate the FDR.

### RNA-Seq analysis

RNA-Seq reads were trimmed with trim galore version 0.4.0 using cutadapt 1.8.1 [38] with flags –q 30 and –illumina. RNA-Seq reads were mapped to the annotated metagenomic assembly using bowtie version 1.1.1 [39]. Candidate structured RNAs as well as those present in Rfam [13] were identified on these assemblies using cmsearch [14] ($E$ value $< 1 \times 10^{-6}$). The number of reads mapping to each structure was calculated using bedtools [35]. If an RNA-Seq read mapped to any position that overlapped with a predicted structured RNA (even at a single position), the read was counted. RPKM values for candidate structured RNAs were calculated based on read counts from all candidate and known structured RNAs. IGV [40] was used to visualize coverage.

### Predicting small genes in HMP2

We predicted small genes using SmORFinder [19] with default settings on HMP2 [9] contigs. We then used bedtools [35] to determine which candidate structured RNAs may overlap with predicted small genes.

### Length of candidate structured RNAs

The lengths of candidate structured RNAs were reported based on the length given in the calibrated cm file for each structure. This represents the maximum length of the structure.

### Taxonomic classification

The contigs in which predictions were identified were classified using One Codex [16]. For every candidate structured RNA, we counted the number of contigs that classified to each taxon at the levels of domain, phylum, class, order, family, genus, and species. We also provide the average percent of k-mers that were classifiable for each candidate structured RNA.

### Genomic neighborhood analysis

To determine the genes that are present within the vicinity of candidate structured RNAs, amino acid sequences of genes that existed within 5 kb of each predicted structured RNA were identified. We then compared these sequences against the Conserved Domain Database (CDD) [41], using RPS-blast [15]. A hit was considered significant if: $E$ value $\leq 0.05$ and the protein aligns to at least 80% of the PSSM's length. To identify protein domains directly downstream of candidate structured RNAs, we performed the

same analysis as above except only considered genes that were within 25 base pairs downstream of candidate structured RNAs.

## Supplementary Information

---

**Additional file 1: Table S1.** Overview of candidate structured RNAs. We show information the 3161 candidate structured RNAs, including the p score from RNAphylo [2], number of significant covarying bases from R-scape [11], length of structure, number of instances of structure in HMP2 [9], taxonomic breakdown of contigs in which the structure is found (including domain, phylum, class, order, family, genus, and species) from One Codex [16], average percent of k-mers mapped for classification, and protein domains found within 5 kb of the structures. In a separate tab, we also include the same analyses for the 2022 candidate structured RNAs that overlapped the nr database. Additionally, we indicate which structures in Rfam [13] were predicted using this pipeline and consequently discarded because they were known structures.

**Additional file 2: Table S2.** Candidate structured RNAs that may overlap with small proteins. We provide a table of candidate structured RNAs that overlap small protein families predicted using SmORFinder [19] and the sequence of the small proteins.

**Additional file 3: Table S3.** Candidate structured RNAs that overlap or are palindromic. We list which candidate structured RNAs are found on the opposing strand of a different candidate structured RNA. All pairwise overlaps are provided both for the 3161 and 2022 (those that overlap the nr database) candidate structured RNAs as separate tabs. Additionally, we list which structures are palindromic, or those in which cmsearch [14] predicts the same structure overlapping the same region in opposing orientations for the 3161 and 2022 candidate structured RNAs as separate tabs.

**Additional file 4: Table S4.** Candidate structured RNAs that may be *cis*-regulatory based on genomic location. We determined which candidate structured RNAs were found directly upstream of genes, specifically within 25 bp of the 5′ of genes. For those genes only, we determined which protein domains were present. These candidate structured RNAs potentially represent those found in potential 5′ UTRs or at the least may be *cis*-acting.

**Additional file 5: Table S5.** RPKM quantifications of structured RNAs. For all candidate structured RNAs in the genomic assemblies from our previous study [29], we provided the RNA-Seq RPKM values of those expressed as well as their genomic position on the assemblies.

**Additional file 6.** Visualization of the 3161 candidate structured RNAs. We show R2R [36] visualizations of the 3161 candidate structured RNAs as a pdf.

**Additional file 7.** Visualization of the set of 2022 candidate structured RNAs that overlap the nr database. We show R2R [36] visualizations of the 2022 candidate structured RNAs with significant hits to the nr database as a pdf.

**Additional file 8.** Alignment files of the 3161 candidate structured RNAs. We provide alignment files for each of the 3161 candidate structured RNAs as well as the calibrated cm file that can be used to search for them.

**Additional file 9.** Alignment files of the 2022 candidate structured RNAs that overlap the nr database. We provide alignment files for each of the 2022 candidate structured RNAs as well as the calibrated cm file that can be used to search for them.

**Additional file 10.** BLASTx results of the 2022 candidate structured RNAs to the nr database. We display the BLASTx [15] results in output format 6 to show with which genes these candidate structures may overlap.

**Additional file 11.** Review history.

---

### Acknowledgements
The authors would like to thank Tessa M. Andermann, Ekaterina Tkachenko, and Joyce B. Kang for major contributions to the fecal biobank of blood and marrow transplantation patients at Stanford hospital, especially Dr. David Miklos who oversees the biobanking protocol and Dr. Andrew Rezvani. We would like to thank the physicians, patients, and nurses involved in collection. We thank Christina Wyss-Coray for collecting samples from patients with Alzheimer's disease, some of which were used in this study. We appreciate sample collection feedback from Victor Henderson and Tony Wyss-Coray. Additionally, we would like to thank Rhiju Das, Aravind Natarajan, and Matthew Grieshop for their helpful discussions and comments on the manuscript. Finally, we would like to gratefully acknowledge the anonymous reviewers, whose helpful and detailed comments enabled us to substantially improve this analysis and manuscript.

### Peer review information

### Review history
The review history is available as Additional file 11.

### Authors' contributions
BJF and ASB conceived of the study. BJF developed the workflow to predict candidate structured RNAs from intergenic regions. BJF and ASB wrote the manuscript. The authors read and approved the final manuscript.

## Funding
Computing costs were supported via NIH S10 Shared Instrumentation Grant (1S10OD02014101), NIH R01 #AI148623-01, and Damon Runyon Clinical Investigator Award to ASB, Stanford ADRC grant # P50AG047366. B.J.F is supported by the National Science Foundation Graduate Research Fellowship DGE-114747 and the Stanford Center for Computation, Evolutionary, and Human Genomics fellowship.

## Availability of data and materials
All data used in this study are publicly available. Contigs from the 1773 HMP2 metagenomes containing at least 5 Mbp of total contig sequence were downloaded from https://www.hmpdacc.org/hmasm2. The samples used for the RNA-Seq analysis can be found under BioProject accession no. PRJNA510123 [29].

# Declarations

## Ethics approval and consent to participate
Not applicable.

## Competing interests
The authors declare no competing interests.

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

## 
