## [**Additional file 11.** Review history. · Genome Biology]

Review History

First round of review Reviewer 1

Are you able to assess all statistics in the manuscript, including the appropriateness of statistical tests used? Yes, and I have assessed the statistics in my report.

Comments to author:

This paper uses transcripts in RNA-seq experiments to detect candidate RNAs, and attempts to validate them using comparative approaches. The RNA-seq data used is four human gut metatranscriptomes. Their respective metagenomes are used for mapping the reads. The candidate transcripts are filtered with various methods, including a supposed test for conserved secondary structure. The paper's main specific predictions are a set of 4 putative tracrRNAs and the detection of RNAs that are antisense to other predicted RNAs.

In my opinion, there is a core of this paper that is absolutely worthy of publishing. However, the claim of structured RNAs is not adequately supported. Moreover, I find other significant issues with the analysis. For these reasons (as I detail below), the paper cannot be published in its present form.

I am uncertain about whether a revised paper would have an appropriate impact for Genome Biology, especially if the structural analysis is not profoundly improved. Many papers have searched for RNAs based on RNA-seq data. I am also aware of one that uses RNA-seq data from a metatranscriptome: (Shi, et al., 2009) (<https://www.nature.com/articles/nature08055>). This paper also looks for sequence conservation and evaluates structural conservation. Unless the results of the paper under review can have much greater confidence, it doesn't seem like it adds enough for Genome Biology.

To my mind, the most interesting finding in the paper is the 40 pairs of RNAs that are in an antisense orientation. Most work on RNAs in an antisense orientation relate to RNAs that are antisense to a gene, as the paper points out. Although one exception is known (also mentioned in the paper), the paper suggests that this could be a more common phenomenon than currently appreciated. Additionally, this aspect of the paper is independent of its claims about conserved structures. I find the claims of new tracrRNAs also plausible and potentially of interest. One possibility would be to focus the paper more on these results.

MAJOR COMMENTS -- STRUCTURAL ANALYSIS

The analysis of structural conservation is inadequate, and does not justify the claim that the candidates have a conserved secondary structure. It seems that the paper simply used CMfinder predictions, and rejected candidates for which CMfinder made no prediction. Is this correct?

CMfinder's false positive rate has not been calculated, and is likely quite high. Moreover, most of the candidate RNAs in the supplementary data exhibit no covariation at all -- in most alignments, nucleotides proposed to base pair don't vary at all. Given these facts, I think the predicted structures are likely to be of poor quality and largely wrong, and most of the predicted structured RNAs likely do not actually conserve a structure.

It is therefore not reasonable for the paper to claim that it has found structured RNAs. The structure predictions are so speculative as to be of uncertain value.

I see 3 ways to deal with this issue:

(1) Remove claims of structure from the paper.

(2) Do significantly more work to validate the structural analysis and estimate a false discovery rate along the lines of this paper: <https://pubmed.ncbi.nlm.nih.gov/30537930/>

(3) Do significantly more work to perform detailed analysis of the 1089 candidates for covariation, along the lines of ref.

2. (Note that the mere use of R-scape to do this is inadequate, because CMfinder will in some cases align sequences so as to find structural conservation that might be invalid, but can fool R-scape. The use of other automated scoring methods would also require more analysis, as in the Kirsch, et al. paper.)

MAJOR COMMENTS -- OTHER

While reading through abstract, introduction and results, I compiled a growing list of problems and caveats with the analysis in this paper. Finally in Discussion, the paper conceded many of these issues. These issues mostly had to do with the limitation that the method can only find RNAs that are expressed in the sample and that that predictions of the taxonomy of metatranscriptome sequences is likely to be imprecise or misclassified. Most of these caveats should have been mentioned much earlier in the paper. A casual reader of the paper could get an unrealistically positive impression of the work, and thus it is important to

mention these caveats earlier. Moreover, the paper does mention positive interpretations before Discussion, so it can also mention negative interpretations.

For example, in Introduction, the paper rightly criticizes previously conducted comparative approaches to discovering novel structured RNAs ("As a consequence of computational limitations and subsetting of intergenic regions, much of the intergenic search space in microbes remains unexplored and many new structured RNAs likely remain to be discovered."). However, the paper fails to mention that it uses essentially the same "subsetting" strategy (requiring transcribed RNAs), and suffers from a similar problem (this requirement for robustly transcribed RNAs was mentioned in Discussion, but not in Introduction).

Also in Introduction, the paper is correct that previous comparative approaches cannot rule out the possibility that their candidate RNAs are in fact ssDNAs. However, given that extremely few ssDNAs with a conserved structure are known, this is a highly theoretical limitation. I think this is a weak point that actually undermines the paper's case. I would recommend that the paper focus on the fact that confirmation of transcription adds evidence to a predicted structured RNA. (It does not, however, confirm that it has a biologically important structure at all.)

Did the paper use an e-value threshold of 100,000 in the cmsearch analysis? If so, this threshold is quite high. I would expect most known RNAs would be found in such a search, but there might be a high number of false positive matches.

(a) This issue calls into question the claims of discovery of DsrA and Spot_42 in other phyla (page 7, line 52 -- page 8, line 18). What is the actual evidence to support the claim that the candidate RNAs are diverged examples of these RNAs? A high e-value is not adequate.

(b) The issue also undermines the statement that 128 of the original 1213 candidates match known RNAs, which is the only evidence that speaks to the accuracy of the paper's candidates.

(c) If the authors decide to use a more stringent e-value threshold, they should also use the score thresholds that have been determined by the Rfam curators for each family (the #=GF GA threshold).

As in Discussion, taxonomy predictions can be incorrect. Since some of the paper's claims depend on accurate taxonomy predictions, the paper needs some more objective statement on what kind of false positives can be expected in the taxonomy predictions, and what evidence supports the claim. I'm looking for a clearer statement, and probably a citation. In addition, for the motifs that are claimed to be homologs of dsrA or spot 42, how accurate is the phylum prediction likely to be in these specific examples, given the specific contigs involved?

Is there any information on the reliability of RNA-seq data, especially in a metatranscriptome context? Is it possible that there are false-positive signals, or meaningless background transcription? If there is no information on this, I think it would be sufficient to mention these issues as factors to consider of uncertain magnitude. (To be explicit, I think actually determining such information would be out of scope of the paper under review.)

The paper should mention the possibility of short ORFs. Although the paper uses RNACode, short ORFs might not yield statistically significant signals. It might be helpful to extend alignments somewhat beyond the apparent RNA-seq signal to capture additional potential codons. Given the lack of positive evidence of structure, the possibility of short ORFs remains significant and should be mentioned.

MINOR COMMENTS

The "cmsearch" program is incorrectly written and (more importantly) incorrectly cited throughout most of the paper. The cited paper for "CMsearch" relates to proteins, and I assume it was not used. "cmsearch" is a part of the Infernal package, for which the correct citation is once cited as ref. 41.

Figure 2A has errors with some of the numbers, which start with commas.

If the paper continues to focus on structured RNAs, a definition of "structured RNA" would be helpful. I assume this means an RNA that has a conserved secondary structure.

The paper found 9,261 intergenic regions with RNA-seq signal and there were 13,377 conserved regions among them. Why did some of the intergenic regions contain more than one conserved region? I think this might occur if BLAST found distinct regions, but it'd be good to explicitly state the reason in the paper. Also, how many intergenic regions had at least one conserved region?

The paper states that the most highly expressed candidate RNA was at 887,778 RPKM. How does this compare to known RNAs, e.g., rRNAs, within the samples? Is this a lot, or a moderate amount, or what?

In the sentence "We found these 1,085 structured RNAs 4,351 times in the Samples A-D in total.", do the 4351 occurrences include the original 1085 occurrences, or are they additional to them?

Figure references are incorrect. For example, the reference to "Figure 2A-D" should be to Fig. 3A-D. There's other incorrect figure references throughout the paper.

The only information relevant to the accuracy of the method is that 128 of the 1213 candidates (~10%) overlapped known RNAs. How much did the different filtering steps affect this precision estimate? How many of the 157995 total intergenic regions contain known RNAs, i.e., how many known RNA overlaps are expected from 1213 random candidates?

What does it mean that DsrA is "more specific" to Proteobacteria? Is it only in Proteobacteria? Is it mostly in Proteobacteria, but also some other taxa? Similarity Spot_42 is "quite specific" to Proteobacteria -- what does that mean?

In Methods, what does "with the following restrictions to include" mean? What was included and what was excluded? Also, why not use the entire Rfam Database? There's also a possibility of non-bacterial content in a metatranscriptome.

When the paper says that tRNAs from Aragorn and rRNAs from Barnnap were not included in this search, what search does this refer to? Were candidates overlapping tRNAs and rRNAs from these sources considered as previously known or not?

What was the commandline used for BLASTN / what non-default settings were used?

In "Taxonomic Classification of Technologies", delete "of Technologies". Otherwise, what technologies were classified?

I don't understand the sentence "In separate searches after our Prokka predictions were generated, we then searched for the Rfam database against these predictions to determine overlap." I assume that the candidate RNAs were searched, but this sentence needs to be edited to make that clear. Also, how were Prokka's predictions of protein-coding genes / intergenic regions used?

"Genomic Neighborhood Analysis of Small Protein Families": should "of Small Protein Families" be deleted? Otherwise, in what way are the families "small", and what do small families have to do with the analysis?

The red & black lines in Figure 4 should be explained.

How was the "Length of structured RNA (bp)" in Fig. 2B calculated? Is this based on the RNA-seq data, or somehow based on the putative structure?

In Fig. 2C, how significant is the difference, given that the novel candidate RNAs were selected for having an RPKM of at least 20? Even ignoring this potential for bias, I'm not really sure what I'm supposed to conclude based on this sub-figure. The text implies that it just wants to say that the novel candidates are expressed, but then there are multiple other aspects to the graph. Also, the format of the box plot should be explained -- what do the various lines & circles mean?

Nucleotides in Fig. 4B overlap. It's difficult to read.

What's the distinction between the candidates in the two worksheets of Table S1: what's the difference between "known structures" and "similar structures"?

Reviewer 2

Are you able to assess all statistics in the manuscript, including the appropriateness of statistical tests used? Yes, and I have assessed the statistics in my report.

Comments to author:

Fremin and Bhatt introduce a novel and original approach to detecting candidate structured ncRNAs in microbes. They follow a discovery pipeline very similar to prior works in the field (notably, riboswitch discovery via comparative genomics by the Braker lab), but use two new information sources to guide their search. Instead of considering specific genomic locations, such as UTRs, where non-coding regulatory RNA elements are anticipated and typically searched for, they use RNA-seq signals from intergenic regions. This allows them to narrow down the search space and make it computationally feasible but at the same time to also focus the search on previously unexplored intergenic regions. The second novelty lies in leveraging human microbiome samples and meta-genome data to expand the repertoire of organisms that may provide evidence of sequence and/or structure conservation. This evidence is important and commonly used to strongly support the "structuredness" (and potentially also functionality) of the identified RNA elements. The new approach leads to several new findings, including the identification of new tracrRNAs and the proposal of many new ncRNA candidates.

This manuscript is well written and well organized. I especially appreciated the detailed description of the various components of

the work (both experimental and computational) and the detailed supporting information the authors provide. I also appreciated the explicit discussion of their method's limitations at the end, which is important but unfortunately missing in many publications. However, a few things were unclear to me and I am not fully convinced that the search for overlap/similarity between Rfam models and the newly obtained models could not have been more stringent. Related questions and comments are below.

1. After refining the set of candidates obtained from the human microbiome data, the authors search for overlaps with structure families already listed in Rfam. My understanding is that this was done by comparing the covariance models the authors obtained (from HMP data) to covariance models in Rfam, which were mined from different DNA sequences / organisms. However, did the authors look into the locations of the hits of their covariance models in the Rfam data vs the locations of hits of Rfam models? In other words, is it possible that CMsearch deems some CMs different yet they predict structures in the same or overlapping genomic regions? More generally, it isn't clear to me where (within intergenic regions) the novel candidates were found and if these locations do not come up when searched by Rfam models. For example, was anything found in UTRs or are the findings located mostly in regions where no ncRNA candidates were previously identified?
2. In that context, how were the tracrRNAs shown in Fig. 4 determined to be significantly different than Rfam models? For example, the structures shown in Fig. 4A-B do bear some resemblance to the R-scape structure shown in RF02348; and the structures shown in Fig. 4C-D do not seem vastly different than the Rfam model for RF02348. My point is: is it possible that extending the search to many new organisms should lead to refinement of some of Rfam's existing models rather than to new models, in particular when genomic locations are similar?
3. I also noticed that the authors report that according to CMsearch, 120 of their candidates do share similarity with Rfam models, but that these were found in different taxa and genomic contexts than the Rfam ones. However, I believe the examples provided in the main text refer to different taxa and not different genomic contexts. Could the authors provide more information and examples on the differing genomic contexts?
4. A relatively recent strategy for discovering structured non-coding motifs makes use of structure probing data, which can be advantageous when searching for relatively small or less conserved motifs (compared to, for example, riboswitches). See Mostoe et al., *Cell*, 2018. This study limited the search to mRNAs and thus identified new candidates in UTRs. If the authors did find candidates in UTRs, it may be worth comparing these findings to Mostoe et al.'s findings or otherwise search Mostoe et al.'s *E. coli* mRNA data for further "structural" support of their findings.

Minor comment: I didn't find any reference to Figure 5 in the main text.

Reviewer reports:

Reviewer #1: This paper uses transcripts in RNA-seq experiments to detect candidate RNAs, and attempts to validate them using comparative approaches. The RNA-seq data used is four human gut metatranscriptomes. Their respective metagenomes are used for mapping the reads. The candidate transcripts are filtered with various methods, including a supposed test for conserved secondary structure. The paper's main specific predictions are a set of 4 putative tracrRNAs and the detection of RNAs that are antisense to other predicted RNAs.

In my opinion, there is a core of this paper that is absolutely worthy of publishing. However, the claim of structured RNAs is not adequately supported. Moreover, I find other significant issues with the analysis. For these reasons (as I detail below), the paper cannot be published in its present form.

I am uncertain about whether a revised paper would have an appropriate impact for Genome Biology, especially if the structural analysis is not profoundly improved. Many papers have searched for RNAs based on RNA-seq data. I am also aware of one that uses RNA-seq data from a metatranscriptome: (Shi, et al., 2009) (<https://www.nature.com/articles/nature08055>). This paper also looks for sequence conservation and evaluates structural conservation. Unless the results of the paper under review can have much greater confidence, it doesn't seem like it adds enough for Genome Biology.

To my mind, the most interesting finding in the paper is the 40 pairs of RNAs that are in an antisense orientation. Most work on RNAs in an antisense orientation relate to RNAs that are antisense to a gene, as the paper points out. Although one exception is known (also mentioned in the paper), the paper suggests that this could be a more common phenomenon than currently appreciated. Additionally, this aspect of the paper is independent of its claims about conserved structures. I find the claims of new tracrRNAs also plausible and potentially of interest. One possibility would be to focus the paper more on these results.

First, we would like to thank you for this extremely thoughtful and helpful review. As this is the first manuscript we have prepared in the structured RNA space, we greatly benefitted from the insight and advice provided by this reviewer. There were many considerations brought up by this review that have enabled us to dramatically improve our work. We thank you for thinking that at its core, this work is worthy of publishing. We hope that the substantial revisions we have made in response allow the manuscript to realize this potential.

We provide a point by point response, below and we also summarize the major changes we made here:

1) We realized that we needed to include more diverse sequences in our analysis as the observation of covariation in structures needed to be stronger in our final output to enhance our confidence that these were, indeed, bona fide structures. To achieve this, we made the

following adjustments in our approach: Instead of initially subsetting intergenic regions that were transcribed and blasting the HMP2 metagenomic contigs against these, we instead performed an all versus all blast using all intergenic regions in the HMP2 metagenomic contigs as well as our samples with RNA-seq data (totalling 214,794,089 intergenic regions above 30 bp in length). While we had considered doing this previously, we found that this would be computationally limited using blast (which we estimated would take 720,000 CPU-hrs); herein, we describe an approach that achieves that objective with a somewhat streamlined computational approach. Specifically, we subsetted the queries and databases into smaller fasta files and performed High speed BLASTn (<https://pubmed.ncbi.nlm.nih.gov/26250111/>) which took roughly 35,000 CPU-hrs at 90GB of RAM. The trade off of this program is fewer CPU-hours necessary at a cost of higher RAM demand. We then clustered these intergenic regions to identify conserved regions. Though we also check to see which of these candidate structures are being expressed (using the RNA-seq data) at the end, we essentially now present an entirely comparative genomics approach for the initial identification of these candidate structured RNAs.

2) We predicted candidate structured RNAs from these new clusters of conserved regions. However, we increased stringency by filtering out any structures that meet the #=GF GA thresholds for known families as well as the previously used e-value cutoff.

3) Additionally, we calculated FDR for these new predictions as described previously (<https://pubmed.ncbi.nlm.nih.gov/30537930/>)

4) From these more confident structured RNA predictions, we then determine taxonomy, protein domains, genomic neighborhoods, and expression.

In summary, we have broadened our approach to identify structured RNAs in all intergenic regions of HMP2 contigs; as HMP2 contains a much broader range of microbial genomic information compared to RefSeq, we feel this approach presents a thorough and broad investigation of structured RNAs in intergenic regions.

MAJOR COMMENTS -- STRUCTURAL ANALYSIS

The analysis of structural conservation is inadequate, and does not justify the claim that the candidates have a conserved secondary structure. It seems that the paper simply used CMfinder predictions, and rejected candidates for which CMfinder made no prediction. Is this correct?

CMfinder's false positive rate has not been calculated, and is likely quite high. Moreover, most of the candidate RNAs in the supplementary data exhibit no covariation at all -- in most alignments, nucleotides proposed to base pair don't vary at all. Given these facts, I think the predicted structures are likely to be of poor quality and largely wrong, and most of the predicted structured RNAs likely do not actually conserve a structure.

It is therefore not reasonable for the paper to claim that it has found structured RNAs. The structure predictions are so speculative as to be of uncertain value.

I see 3 ways to deal with this issue:

- (1) Remove claims of structure from the paper.
- (2) Do significantly more work to validate the structural analysis and estimate a false discovery rate along the lines of this paper: <https://pubmed.ncbi.nlm.nih.gov/30537930/>
- (3) Do significantly more work to perform detailed analysis of the 1089 candidates for covariation, along the lines of ref. 2. (Note that the mere use of R-scape to do this is inadequate, because CMfinder will in some cases align sequences so as to find structural conservation that might be invalid, but can fool R-scape. The use of other automated scoring methods would also require more analysis, as in the Kirsch, et al. paper.)

We entirely agree with all of these points and thank the reviewer for this feedback. Among the three proposed ways to deal with these issues, we felt that option (1), while simplest to implement, was the easy way out but did not really achieve what the paper strived for. Essentially, we decided to implement option (2). However, we reworked the paper from the ground up, starting with an all vs. all HS-BLASTN, to improve covariation and make a stronger case for proposed structures as pointed out in option (3).

MAJOR COMMENTS -- OTHER

While reading through abstract, introduction and results, I compiled a growing list of problems and caveats with the analysis in this paper. Finally in Discussion, the paper conceded many of these issues. These issues mostly had to do with the limitation that the method can only find RNAs that are expressed in the sample and that that predictions of the taxonomy of metatranscriptome sequences is likely to be imprecise or misclassified. Most of these caveats should have been mentioned much earlier in the paper. A casual reader of the paper could get an unrealistically positive impression of the work, and thus it is important to mention these caveats earlier. Moreover, the paper does mention positive interpretations before Discussion, so it can also mention negative interpretations.

For example, in Introduction, the paper rightly criticizes previously conducted comparative approaches to discovering novel structured RNAs ("As a consequence of computational limitations and subsetting of intergenic regions, much of the intergenic search space in microbes remains unexplored and many new structured RNAs likely remain to be discovered."). However, the paper fails to mention that it uses essentially the same "subsetting" strategy (requiring transcribed RNAs), and suffers from a similar problem (this requirement for robustly transcribed RNAs was mentioned in Discussion, but not in Introduction).

Also in Introduction, the paper is correct that previous comparative approaches cannot rule out the possibility that their candidate RNAs are in fact ssDNAs. However, given that extremely few

ssDNAs with a conserved structure are known, this is a highly theoretical limitation. I think this is a weak point that actually undermines the paper's case. I would recommend that the paper focus on the fact that confirmation of transcription adds evidence to a predicted structured RNA. (It does not, however, confirm that it has a biologically important structure at all.)

Did the paper use an e-value threshold of 100,000 in the cmsearch analysis? If so, this threshold is quite high. I would expect most known RNAs would be found in such a search, but there might be a high number of false positive matches.

(a) This issue calls into question the claims of discovery of DsrA and Spot_42 in other phyla (page 7, line 52 -- page 8, line 18). What is the actual evidence to support the claim that the candidate RNAs are diverged examples of these RNAs? A high e-value is not adequate.

(b) The issue also undermines the statement that 128 of the original 1213 candidates match known RNAs, which is the only evidence that speaks to the accuracy of the paper's candidates.

(c) If the authors decide to use a more stringent e-value threshold, they should also use the score thresholds that have been determined by the Rfam curators for each family (the #=GF GA threshold).

Thank you for this suggestion. In this revision, we started from scratch essentially to build entirely new models. Even though these models addressed in your comments are no longer proposed in the paper, we want to still directly address these points:

(a) and (c)

Using the #=GF GA threshold, the structures with similarity to DsrA and Spot_42 were not identified as DsrA or Spot_42. Essentially, these were just high (> 1000) but not significant hits to these structures and also were found in organisms you typically wouldn't see DsrA or Spot_42. However, one of the four new tracrRNAs we proposed actually did meet the #=GF GA threshold of a known tracrRNA; therefore, we only found 3 tracrRNAs from the previous analysis. In our new analyses, none of these were predicted due to much higher stringencies in covariation.

(b)

If we used the #=GF GA threshold, we found that 146 additional proposed structures hit existing Rfam models. So assuming we pushed forward with the previous models using your proposed thresholding (which we did not in the revised version of the manuscript) we would have found that of the 1213 candidates, 274 (~23 percent) were known structured RNAs and 939 were novel candidates. Because covariation was not strong in these models, we ultimately did not decide to take this path and instead built better models. To avoid confusion, we just wanted to restate that we are directly addressing these concerns here even though they no longer apply to the current paper and models presented.

As in Discussion, taxonomy predictions can be incorrect. Since some of the paper's claims depend on accurate taxonomy predictions, the paper needs some more objective statement on what kind of false positives can be expected in the taxonomy predictions, and what evidence supports the claim. I'm looking for a clearer statement, and probably a citation. In addition, for the motifs that are claimed to be homologs of dsrA or spot 42, how accurate is the phylum prediction likely to be in these specific examples, given the specific contigs involved?

Thank you for this comment. In the revised manuscript, we cite the reference used for this analysis and also report the percentage of k-mers classified to a taxa for each structure, which can be interpreted as how confident we are that sequences map to the taxa they are called. This at least gives some insights to how well we are classifying a candidate structure on average and is also what our lab has done in the past with small protein classifications (Sberro et al, 2019 in Cell).

Is there any information on the reliability of RNA-seq data, especially in a metatranscriptome context? Is it possible that there are false-positive signals, or meaningless background transcription? If there is no information on this, I think it would be sufficient to mention these issues as factors to consider of uncertain magnitude. (To be explicit, I think actually determining such information would be out of scope of the paper under review.)

We thank the reviewer for pointing out this limitation, which should be mentioned in the paper. We use a stringent RPKM cutoff of 20 (a moderate expression level) to lower the chances of false-positive signals. Transcription does not imply that the prediction is indeed functional or indeed adopts an RNA structure. For example, UTR regions will be transcribed, will not be coding, and it doesn't imply a structure exists there. We now present these as limitations.

After the revisions, we now build our confidence from comparative genomics and the FDR we assign based on shuffled alignments. We still look at RNA-Seq with an RPKM cutoff, but more so as a method that can highlight potentially more interesting structures in the microbiome.

The paper should mention the possibility of short ORFs. Although the paper uses RNACode, short ORFs might not yield statistically significant signals. It might be helpful to extend alignments somewhat beyond the apparent RNA-seq signal to capture additional potential codons. Given the lack of positive evidence of structure, the possibility of short ORFs remains significant and should be mentioned.

Thank you for bringing up sORFs as a possibility. Indeed, recent work from our lab used RNACode for this very purpose -proposing new small genes in microbiomes (Sberro et al, 2019). You are correct in stating that the false negative rate is likely extremely high in this set of sORFs - those sORFs with insignificant RNACode values may still be real sORFs. In the manuscript, we now explicitly say in the limitations: 'Small genes, for example, are often overlooked and absent in databases. We cannot guarantee if a candidate structure RNA is truly in an intergenic region.'

We believe this is why we truly needed better covariation in models and ultimately decided to reshape the paper emphasizing comparative genomics.

To address this concern, we also included a small gene analysis using a recently published tool from our lab smORFinder (Durrant and Bhatt, 2020). The limitations listed above still remain as the false negative rate of predicting small genes is still likely high. But the results suggested 99 of our structures may overlap small genes (Table).

MINOR COMMENTS

The "cmsearch" program is incorrectly written and (more importantly) incorrectly cited throughout most of the paper. The cited paper for "CMsearch" relates to proteins, and I assume it was not used. "cmsearch" is a part of the Infernal package, for which the correct citation is once cited as ref. 41.

Thank you for pointing out this mistake. We now cite Infernal and write "cmsearch" correctly.

Figure 2A has errors with some of the numbers, which start with commas.

Thank you for catching these errors. We paid close attention to this as we recreated figures with the new models.

If the paper continues to focus on structured RNAs, a definition of "structured RNA" would be helpful. I assume this means an RNA that has a conserved secondary structure.

This is a good point. We now state in our intro: "We define structured RNAs as any non-coding RNA with a conserved secondary structure across taxa"

The paper found 9,261 intergenic regions with RNA-seq signal and there were 13,377 conserved regions among them. Why did some of the intergenic regions contain more than one conserved region? I think this might occur if BLAST found distinct regions, but it'd be good to explicitly state the reason in the paper. Also, how many intergenic regions had at least one conserved region?

Thank you and yes you are correct. Some intergenic regions contained multiple conserved regions. If a region was not fully conserved across the length, it could be split into multiple regions that were considered separately.

Also, we previously were not performing all-versus-all blast or being as strict on what constitutes homology; therefore, we likely weren't clustering homologous regions as effectively as possible, resulting in more relatively, smaller clusters that might have otherwise not existed or merged with another cluster. Now, we ignore clusters of only identical sequences in them and do a better job merging more distant sequences indirectly with all-versus-all blast. In the current

paper, we identified roughly 8 million conserved clusters from roughly 200 million intergenic regions.

The paper states that the most highly expressed candidate RNA was at 887,778 RPKM. How does this compare to known RNAs, e.g., rRNAs, within the samples? Is this a lot, or a moderate amount, or what?

Thank you for asking. Some of the most highly expressed ribosomal RNAs would have also been in the hundreds of thousands of RPKM. In the new manuscript, we are not focusing much on RNA-Seq. Now we mainly use RNA-seq in Figure 5 just to show a signal distribution that seems very specific to the structured RNAs we predict. At the very least, it suggests that some of the motifs exist at an RNA level even though it does not validate their structure.

In the sentence "We found these 1,085 structured RNAs 4,351 times in the Samples A-D in total.", do the 4351 occurrences include the original 1085 occurrences, or are they additional to them?

We agree this wording was confusing. This portion of the manuscript has been substantially revised; for the reviewer's benefit, we answer the question here, though it no longer is relevant to the revised manuscript: We meant that some of 1,085 candidate structured RNAs could be found more than once in the samples. So in other words, this just meant that on average we are finding each of these 1,085 structures ~4 times across the samples. We have dramatically downplayed RNA-Seq in the paper now as we just use it to highlight signal distribution of candidates.

Figure references are incorrect. For example, the reference to "Figure 2A-D" should be to Fig. 3A-D. There's other incorrect figure references throughout the paper.

Thank you for pointing this out. We were more careful in this revision of figures.

The only information relevant to the accuracy of the method is that 128 of the 1213 candidates (~10%) overlapped known RNAs. How much did the different filtering steps affect this precision estimate? How many of the 157995 total intergenic regions contain known RNAs, i.e., how many known RNA overlaps are expected from 1213 random candidates?

We appreciate this feedback. We agree this is not a good way to suggest our approach is accurate as we should have an FDR to specifically address the novel candidates we are proposing. In the revised manuscript, we ensure that every structure has significant covariation (R-scape) and most importantly, we shuffled the alignments and calculated FDR of the structures we are proposing.

What does it mean that DsrA is "more specific" to Proteobacteria? Is it only in Proteobacteria? Is it mostly in Proteobacteria, but also some other taxa? Similarity Spot_42 is "quite specific" to Proteobacteria -- what does that mean?

Thank you for pointing this out. Of note, we no longer make arguments like this as they are very speculative. Also, these models are no longer being proposed in the current manuscript.

However to address the question, we meant that if you look at DsrA in Rfam, you would find that 32 of the 34 species it is found in belong to Proteobacteria. Similarly, Spot 42 is found 184 times in Proteobacteria species out of 192 species in Rfam. It could just be that existing models for these ncRNAs are very specific to proteobacteria even though there may be similar structures performing similar functions in other taxa. Again, the argument of 'almost significant' hits to known structures is not very strong. In the revised manuscript, we no longer address cases like these.

In Methods, what does "with the following restrictions to include" mean? What was included and what was excluded? Also, why not use the entire Rfam Database? There's also a possibility of non-bacterial content in a metatranscriptome.

We appreciate this insight. In fact, we propose candidate structured RNAs for nonbacterial constituents of the microbiome in the revised manuscript. We now use the entire Rfam database.

When the paper says that tRNAs from Aragorn and rRNAs from Barnnap were not included in this search, what search does this refer to? Were candidates overlapping tRNAs and rRNAs from these sources considered as previously known or not?

Thank you for this point. Previously, we were using Aragorn and Barnnap instead of Rfam to predict rRNAs and tRNAs. We then were excluding rRNAs and tRNAs from our cmsearch against Rfam. In the current manuscript, we use Rfam to determine if a model is a tRNA or rRNA by essentially searching all structures in Rfam. Then, in the case of HMP_734, we actually find that sometimes Aragorn proposes the region as a tmRNA, which we then use to propose that HMP_734 is likely a new tmRNA model.

What was the commandline used for BLASTN / what non-default settings were used?

We used default settings. Also, we use default settings for HS-BLASTN in the updated manuscript.

In "Taxonomic Classification of Technologies", delete "of Technologies". Otherwise, what technologies were classified?

Thank you. This has been deleted.

I don't understand the sentence "In separate searches after our Prokka predictions were generated, we then searched for the Rfam database against these predictions to determine overlap." I assume that the candidate RNAs were searched, but this sentence needs to be edited to make that clear. Also, how were Prokka's predictions of protein-coding genes / intergenic regions used?

Thank you for pointing out this confusing sentence. Yes we meant the candidate RNAs were searched against Rfam. Our approach is a bit different now.

We previously used Prokka (which uses Prodigal to predict genes) to determine which regions were intergenic. Then regions that were not considered coding were defined as intergenic. However, prokka also runs several other tools to predict structured RNA, even including a partial cmsearch against rfam. This made it very confusing and also not ideal.

Now we use Prodigal across all of HMP2 to find what is protein coding. Then we use the regions that were not called coding for downstream analysis. Once we identify possible structured RNAs, we then searched all of Rfam to determine which structures were already known. Then later, we also run Prokka mostly just to see if other tools like Aragorn might have, for example, predicted tRNAs that are not in rfam. With Prokka, we realized that HMP2_734 is likely a tmRNA and HMP2_13009 is likely a repeat in a CRISPR array.

"Genomic Neighborhood Analysis of Small Protein Families": should "of Small Protein Families" be deleted? Otherwise, in what way are the families "small", and what do small families have to do with the analysis?

Thank you for catching this error. We deleted this.

The red & black lines in Figure 4 should be explained.

We appreciate this. Figure 5 in the new manuscript shows genes and structures in a similar way, as we explain these now.

How was the "Length of structured RNA (bp)" in Fig. 2B calculated? Is this based on the RNA-seq data, or somehow based on the putative structure?

We appreciate this point. We added this to our methods. We calculated lengths based on the length reported in the calibrated cm files for each structure (which we now provide with the alignments in the supplemental files 3 and 4). This length is the maximum length of the structure when all bases are present.

In Fig. 2C, how significant is the difference, given that the novel candidate RNAs were selected for having an RPKM of at least 20? Even ignoring this potential for bias, I'm not really sure what I'm supposed to conclude based on this sub-figure. The text implies that it just wants to say that the novel candidates are expressed, but then there are multiple other aspects to the graph. Also, the format of the box plot should be explained -- what do the various lines & circles mean?

Thank you for this comment. We agree that we are creating a bias because every candidate structured RNA had at least one instance in which it was RPKM > 20. We have downplayed RNA-seq and removed this from the manuscript.

Nucleotides in Fig. 4B overlap. It's difficult to read.

This model does not exist in our analysis anymore. We took more care to select models that rendered more clearly with R2R.

What's the distinction between the candidates in the two worksheets of Table S1: what's the difference between "known structures" and "similar structures"?

We removed this analysis from the paper. However, we meant that 'known structures' were those with a strong E value (>100000) and those that were 'similar structures' had a weaker E value (> 1000 but < 100000). We chose a strict E value but also wanted to include hits that, although did not meet the cutoff, were still similar in structure. Now, we require a GF GA threshold and E value >100000 for known structures and do not propose similar structures.

Reviewer #2: Fremin and Bhatt introduce a novel and original approach to detecting candidate structured ncRNAs in microbes. They follow a discovery pipeline very similar to prior works in the field (notably, riboswitch discovery via comparative genomics by the Braker lab), but use two new information sources to guide their search. Instead of considering specific genomic locations, such as UTRs, where non-coding regulatory RNA elements are anticipated and typically searched for, they use RNA-seq signals from intergenic regions. This allows them to narrow down the search space and make it computationally feasible but at the same time to also focus the search on previously unexplored intergenic regions. The second novelty lies in leveraging human microbiome samples and meta-genome data to expand the repertoire of organisms that may provide evidence of sequence and/or structure conservation. This evidence is important and commonly used to strongly support the "structuredness" (and potentially also functionality) of the identified RNA elements. The new approach leads to several new findings, including the identification of new tracrRNAs and the proposal of many new ncRNA candidates.

This manuscript is well written and well organized. I especially appreciated the detailed description of the various components of the work (both experimental and computational) and

the detailed supporting information the authors provide. I also appreciated the explicit discussion of their method's limitations at the end, which is important but unfortunately missing in many publications. However, a few things were unclear to me and I am not fully convinced that the search for overlap/similarity between Rfam models and the newly obtained models could not have been more stringent. Related questions and comments are below.

We thank the reviewer for their kind words on the organization, writing, and the way we addressed limitations. We also appreciate the reviewer's appropriate concerns about the level of stringency applied to deem a model novel. These concerns are well warranted. In the first version of this manuscript, we performed the analysis with a specific BLASTn to transcribed intergenic regions, and thus we pulled in less diversity in sequences for each structure. This approach limited diversity and covariation for the structures, which significantly hinders our ability to reliably identify truly new structured RNA models. In the revised manuscript, we downplay the importance of RNA-Seq and instead use significant computational resources to perform an all versus all HS-BLASTN on all predicated intergenic regions in the HMP2 contigs.

The original version of the manuscript focused on using RNA-Seq to lower computational burden associated with studying all intergenic regions, whereas the current manuscript overcomes this computational burden by using modified blast tools (which are faster) to perform a very computationally expensive analysis across all intergenic regions in HMP2. Also with this current approach, we build both better candidate structured RNA models and also have a much greater search space to determine if any of the candidate structured RNAs ever hit the same regions in HMP2 as known structured RNAs in Rfam. In the revised manuscript, we make a stronger case that the candidate structured RNAs hit distinct regions compared to those of known structured RNAs, requiring that our candidate structured RNAs demonstrate strong hits to unique regions.

1. After refining the set of candidates obtained from the human microbiome data, the authors search for overlaps with structure families already listed in Rfam. My understanding is that this was done by comparing the covariance models the authors obtained (from HMP data) to covariance models in Rfam, which were mined from different DNA sequences / organisms. However, did the authors look into the locations of the hits of their covariance models in the Rfam data vs the locations of hits of Rfam models? In other words, is it possible that CMsearch deems some CMs different yet they predict structures in the same or overlapping genomic regions? More generally, it isn't clear to me where (within intergenic regions) the novel candidates were found and if these locations do not come up when searched by Rfam models. For example, was anything found in UTRs or are the findings located mostly in regions where no ncRNA candidates were previously identified?

Thank you for this comment. We agree that this was a confusing part of the manuscript because it was less linear than it should have been. We previously searched four fecal sample metagenomic assemblies for our (1) candidate structured RNAs (using the CMs we generated) and (2) all the CMs in Rfam. Then using bedtools, we determined if any of the candidate

structured RNAs ever overlapped with any of the known structured RNAs. If so, they were discarded. However, we did not explain this as well as we should have. In the current manuscript, we hopefully make this more linear.

In the revised manuscript, we search HMP2 intergenic regions against both Rfam CMs and the candidate structured RNAs we provide. We then discard any candidates with overlap to known structures in Rfam using (1) a GF GA threshold and (2) an e-value threshold (> 1000000). This is arguably much better because HMP2 has a much greater variety of sequences than just the four fecal samples we searched initially. As an example, let us consider a candidate structure that is found 100 times in HMP2. If even one of those times, it overlaps, even partially, a known structure in Rfam, it is no longer considered a novel candidate.

Finally, the reviewer brings up a good point about UTRs. Using available knowledge and tools, it is very difficult to definitively determine what is a UTR and what is not. However, we do perform an analysis (Table S4) that shows which protein domains are found in genes that are immediately downstream (within 25 bp) of structured RNAs. These are structures that are potentially in the 5' UTR.

2. In that context, how were the tracrRNAs shown in Fig. 4 determined to be significantly different than Rfam models? For example, the structures shown in Fig. 4A-B do bear some resemblance to the R-scape structure shown in RF02348; and the structures shown in Fig. 4C-D do not seem vastly different than the Rfam model for RF02348. My point is: is it possible that extending the search to many new organisms should lead to refinement of some of Rfam's existing models rather than to new models, in particular when genomic locations are similar?

Thank you for this point. You are correct in this assessment. In our new analyses, none of these models were predicted due to much higher stringencies in covariation.

However to address this, one of the four new tracrRNAs we previously proposed actually did meet the $\#GF GA$ threshold of a known tracrRNA (even though the e-value did not meet our thresholds); therefore, we technically would have only found 3 new tracrRNA models from the previous analysis if we were more stringent. There is a general issue that seems to occur in which structures may be very similar to others but not quite close enough to meet cutoffs to be called the same structure. Many times, this likely occurs because the same structure may be present in different taxa we haven't inspected before and the structure in that taxon has enough differences to not be captured by an existing model.

Though there is a plethora of grey area and distinctions in cutoffs for related structures, the overall goal here is to be able to uniquely and accurately capture structured RNAs in HMP2, even if it means sometimes building similar models better suited to the taxa we haven't inspected before. As long as the two similar models do not strongly hit the same regions, this still adds value despite being imperfect.

Attempting to combine multiple models (by aligning all the sequences of potentially related models and trying to recreate a larger model) into a more comprehensive one can also be tricky as it can be difficult to know if combining is actually the better alternative than having multiple more specific models. Also it is difficult to know what cutoffs to use to suggest when two models should be merged into one. This likely varies on a case-by-case basis and is hard to define what is better. One of many limitations of our work is that there are some candidate structured RNAs that we are discarding because sometimes they overlap a region also hit by an Rfam model even though that Rfam model may not be doing a good job capturing all the possibilities. Though we are not sure what the best approach would be in trying to improve existing Rfam models and quantify how much better or worse they are afterwards, we definitely agree this is a challenging limitation to address in the future. This might require searching many more sequences than HMP2 and coming up with some defensible thresholds on which models should be combined if any overlap in regions is observed.

3. I also noticed that the authors report that according to CMsearch, 120 of their candidates do share similarity with Rfam models, but that these were found in different taxa and genomic contexts than the Rfam ones. However, I believe the examples provided in the main text refer to different taxa and not different genomic contexts. Could the authors provide more information and examples on the differing genomic contexts?

Thank you for pointing this out. We no longer have this analysis in the manuscript because it is speculative and based on "almost significant" E values. However, to answer the reviewer's thoughtful question: we meant here that we performed a genomic neighborhood analysis looking at protein domains present within 5 kb of each structure. We found that many of the structures that are similar to known were both found in different taxa and also near different protein domains. In the current manuscript, we make better use of our genomic neighborhood analysis but try to avoid speculative comments like these.

4. A relatively recent strategy for discovering structured non-coding motifs makes use of structure probing data, which can be advantageous when searching for relatively small or less conserved motifs (compared to, for example, riboswitches). See Mostoe et al., Cell, 2018. This study limited the search to mRNAs and thus identified new candidates in UTRs. If the authors did find candidates in UTRs, it may be worth comparing these findings to Mostoe et al.'s findings or otherwise search Mostoe et al.'s E. coli mRNA data for further "structural" support of their findings.

Thank you for this comment. This is a great idea. We followed up on this but unfortunately, there are only three candidate structured RNAs in the genus Escherichia in our resource. We were not able to find any of these in Mostoe et al. though it would have been a nice way to both provide structure support and a model for these structures.

Minor comment: I didn't find any reference to Figure 5 in the main text.

We thank the reviewer for this comment. This was a numbering error. As we restructured the paper, we took care to avoid this.

Second round of review

Reviewer 1

This paper is vastly improved in comparison to the previous version. I commend the authors on their response! I'm fundamentally in support of this paper being ready for publication. However, there's a few claims related to the FDR estimates that I think are not entirely justified. I think these issues can be addressed with some small edits to the text, as the analysis in the paper seems fundamentally reasonable to me for a publication. Additionally, I also have a list of further minor points that I think can also be addressed by simple rewrites (except for a potential bug in the drawings, which however does not seem to affect the underlying analysis).

As the Discussion essentially states, the results in this paper are a first step towards finding RNAs in these locations. I think that a significant amount of work will be needed to actually find good candidates for experiments, and that the true FDR is likely much higher than what was estimated, as I explain below. However, I think the analysis performed in this paper is well done and is a meaningful step towards this goal. Moreover, in my opinion, the paper makes a number of interesting choices in which tools to use and how to combine results that I think will be of interest to others.

ABOUT THE FDR ESTIMATES

The paper now does an overall very reasonable and quite thoughtful job of an analysis of FDR with the SISSIz tool. However, I think it's a bit simplistic to assume that the estimated FDR is close to the true FDR. The main issue is that the shuffled alignments do not take into account sources of false positives due to other biological factors that look somewhat like RNAs. These other factors can bias alignments to look more like RNAs in a way that the SISSIz model doesn't take into account. These factors include palindromic binding sites of dimeric proteins, Rho-independent terminators, repeats & remnants of repeats, etc. It is not, of course, feasible to fully model all such features -- many of them are probably not even yet known to science. So, I think the paper's analysis is fundamentally good. But the FDR estimate is nonetheless not likely to be reliable.

A arbitrarily looked at the first few predicted candidates (by ID number), and most seem dubious to me:

- HMP2_1 : this looks like a Rho-independent terminator (so, it is at least an RNA, but there's no evidence of a new RNA type)
- HMP2_2 : The first 3 sequences look like another Rho-independent terminator, and the 4th and 5th sequences do not seem homologous (which negates considerations of covariation, as covariation is meaningless without a homology relationship)
- HMP2_9: the first stem does not exhibit covariation, although it does look somewhat like a terminator. The second stem is highly erratic, is missing in many sequences, and the little covariation there is seems to arise from arbitrary shifts in some very A-U-rich sequences.
- HMP2_10: the covariation in the 3' half of the alignment seems to mostly arise from sequences that don't seem homologous. The stem contained in the 5' half of the alignment has more energetically unfavorable base pairs than examples of covariation, although it could be valid.

Anyway, these observations don't seem to suggest an FDR of 2.5-5%.

I also see a possible subtle bias in the way the tests with SISSIz were done, although I don't see a feasible strategy to get around this problem. First, I assume CMfinder produced alignments based on entire intergenic regions, but the re-analysis of the SISSIz shuffled alignments used only the part that was incorporated into the alignment. Moreover, the real CMfinder alignments had an opportunity to look at all intergenic regions to potentially find false positives, whereas the FDR estimates only looked at a much smaller set of sequences (those in alignments chosen by CMfinder and the various filters). It's a bit like if we flipped 2 coins 1000 times, and found that both coins were heads in 250 tosses. Then we try to estimate a background probability for the 250 double-heads by re-flipping the coins in the 250 examples, and get 62 re-flips with both heads -- we should have used the original 1000 tosses. I don't think it's feasible to any better than the current paper's analysis, because analyzing all intergenic regions would require somehow randomizing all metagenome sequences while precisely preserving homology relationships (which seems very hard and is why SISSIz requires an alignment). It also wouldn't solve the problem of confounding biological phenomena. Also, I found it great that the paper repeats the full pipeline on the shuffled alignments, so that they are as least as comparable as possible. So, once again, I think the paper did a good job with the analysis, but this might be an additional reason why the FDR estimates could be underestimated.

To be clear, I think the paper does as well as possible to control for issues, and I'm not suggesting any kind of mistake. (Also, I should say that the paper I recommended in my previous review has analogous problems that aren't discussed in that paper; looking back, I should have mentioned these problems more clearly in my previous review of the current paper.) Nonetheless, these are the limits of the current state-of-the-art, and I think the paper should be more careful in claiming that the predictions are high quality, or that it's likely that there really are only 2.5-5% false positives.

So, after all that, basically I recommend rewording the "high confidence" claims throughout the paper. Something like a

"carefully constructed automated analysis", for example. Perhaps the paper could briefly mention potential problems with the FDR estimates (no need to go into as much detail as I have here), and describe them as a rough lower-bound on the true FDR rate.

Additionally, I find this sentence a problematic because of the difficulties with accurate FDR estimates: "This is a substantial contribution of candidate structured RNA families; for reference, Rfam 14.3 [15] currently contains 3,446 families." Most Rfam families have gone through a much more stringent process than those in the paper under review. (Although, I'll admit that some of the early E. coli RNAs got included in Rfam too easily.) I don't think it's appropriate or meaningful to compare the 3161 predictions to the 3446 families in Rfam.

I should also say that, despite my caveats, I do find the FDR estimates to be useful, as they give some indication that the analysis is likely to be good. Even if they might be a kind of lower bound on the true FDR, I think it's an important control to do this kind of analysis -- if the FDR had been estimated at 90%, we'd know there was some kind of serious problem, whereas the low apparent FDR at least says that many of the predictions are likely to be meaningful. For this reason, I think the revised past is a vast improvement over the previous version (which had no controls), and I'm basically in support of publication.

MINOR COMMENTS RELATED TO COMPUTATIONAL LIMITATIONS

I felt there were a few statements that were a little off related to the claim that the major problem with the basic comparative approach is computational limitations.

The paper makes claims in a few places similar to the following: "While powerful, a drawback of this comparative genomics approach is that in any given experiment, only a select set of intergenic regions are typically considered; this is done to accommodate computational limitations. For example, comparative genomics approaches often subset intergenic regions to bias towards known classes of structured RNAs, like ribozymes or specific types of riboswitches, which tend to be found in specific genomic contexts [2]."

The relevant part of ref [2] reads "However, the complete set of available genomic and metagenomic IGRs totals 37 billion base pairs, which presents two difficulties. First, the computational time and memory required for analysis expands with many IGRs. More crucially, the false positive rate when IGRs are compared increases with their total number, making it difficult to extract rare RNAs."

The paper under review seems to ignore the sentence beginning "more crucially". I would guess that HS-BLASTN has a high specificity and relatively low sensitivity (in order to be fast), and this reduces the false positives in matches. Unfortunately, this presumably also reduces sensitivity. I think the paper under review should also briefly mention the challenge of false positives or false negatives in analyzing huge datasets, which might compromise the ability to actually find RNAs in huge datasets.

(2) The paper under review more or less solves this computational load problem, but doesn't explicitly say this. The first paragraph of Discussion gets at this, even citing ref [2] again, but doesn't discuss what the innovation was, or explicitly say that there was no need to limit intergenic region (except that only human microbiomes were used). My sense is that the major innovation is the use of HS-BLASTN (combined with a lot of CPU hours). Some more direct statement on this in the Discussion would be great.

MISCELLANEOUS MINOR COMMENTS

I'm not 100% clear on how the RNA-seq analysis for RNA candidates was conducted. I think what happens is that the locations of instances of each alignment are compared to regions that are expressed in RNA-seq data. Is this true? What happens if a sequence belonging to the alignment is only partially expressed in the RNA-seq data (e.g., the 5' half overlaps RNA-seq reads, but not the 3' half)? If a large region is expressed according to RNA-seq, and a prediction covers a part of this, I assume that counts as expressed, correct? Basically, I assume that if one nucleotide of an RNA candidate overlaps one nucleotide of an RNA-seq read, then it's considered to be expressed, right? I'm not suggesting any potential problem here, just that it would be nice to be more explicit about this in the paper. (Could also be supplementary text.)

It might be appropriate to cite GraphClust (<https://www.ncbi.nlm.nih.gov/pmc/articles/PMC3371856/>) and/or GraphClust2 (<https://www.ncbi.nlm.nih.gov/pmc/articles/PMC6897289/>) as another approach to clustering large datasets, perhaps with a brief sentence. This program uses locality-sensitive hashing, and the amount of time it takes is theoretically linear in the number of input sequences, as opposed to quadratic with BLAST. To my knowledge, GraphClust has not actually been applied to truly huge datasets, but at least in principle it could be.

The paper analyzes genes within 5 Kb of RNA instances, which as I would understand it, includes genes upstream and downstream. What's the biological motivation for this? I think it's rare that an RNA's function has a strong association with

genes that are nearby in either the 5' or the 3' direction. Also, the statistic fails to take into account the fact that some genes are common, e.g. HTH-XRE. The paper could leave the existing analysis with some kind of rationale, and just point out that it doesn't account for gene frequency. If the paper has a particular goal in citing this statistic, it might be good to normalize the gene frequency near to the RNA by dividing it by the total number of gene occurrences in the dataset. I'd recommend simply removing the analysis entirely.

I don't see a description of how CMfinder was used for the predictions. As I understand it, CMfinder normally produces multiple predictions. What command line was used / how was it processed?

It'd be nice if the paper would report the total number of basepairs in the set of intergenic regions that were analyzed.

The response to review states that it took 35,000 CPU hours at 90 GB RAM to do the HS-BLASTN comparisons. I think this information would be interesting for the paper, and perhaps the other information in the paragraph of the authors' response (why NCBI BLAST wouldn't be feasible, and something like the statement "The trade off of this program is fewer CPU-hours necessary at a cost of higher RAM demand.") I actually read the paper before looking at the responses, and wanted to know more about the resources required by HS-BLASTN, and then saw that was discussed in the response to my review.

What was the rationale for filtering HS-BLASTN matches as described (≥ 30 bp, $< 100\%$ identity, < 100 bit score)? I don't see any need to validate these choices, but a statement about why they were chosen would be nice, even if it was just to say, for example, that it was based on the authors' intuition.

In filtering HS-BLASTN results, was there a minimum score, or some kind of threshold to remove meaningless matches? What was the relevant default setting in HS-BLASTN (in terms of a score or E-value threshold)?

I'm unclear on how the fasta files were split for HS-BLASTN, with this sentence from the manuscript: "we split the intergenic regions into 100 roughly equal smaller fasta files and ran HS-BLASTN [8] on each file before combining to accommodate RAM requirements." First, I think "to accommodate RAM requirements" belongs elsewhere in the sentence (or perhaps this could be broken into more than one sentence). More importantly, I'm not sure I get the idea. Was HS-BLASTN initially used on one fasta file to remove highly similar sequences within that file, so that the total number of sequences is fewer? If so, that would certainly reduce the number of comparisons, but how was it defined whether sequences were too similar (and could be removed)? The stated filters described earlier in the paper (≥ 30 bp, $< 100\%$ identity, < 100 bit score) filter the matches, not sequences. If sequences were not removed from the individual fasta files, then I don't see how processing the individual fasta files would help to reduce computer time. I think there's one or two pieces of information missing here.

"Rfam 14.31" is mentioned in one case. I assume that version 14.3 is meant.

What does the yellow shading in Fig 2D mean?

How was R2R used in this paper? I am surprised by the extremely small number of covarying base pairs in most of the figures within the main text. Moreover, since R-scape was used in the filter, it's surprising if there are really no covarying base pairs at all in some of the candidate RNAs (HMP2_3156 in Fig 3D, HMP2_4078 in Fig 4A, HMP2_1065 in Fig 4C). Looking at the raw alignments, it seems like there should have been covariation according to R2R or R-scape. So it seems like something weird is going on with converting the alignments to drawings.

Fig 2: some nucleotides are difficult to read because they overlap.

Reviewer 2

The authors made substantial changes to their analysis and consequently to the focus of their work and manuscript. Specifically, they addressed the comments and flaws previously pointed out, which led them to design a more robust and reliable pipeline for structured RNA detection via standard comparative genomic analysis methods. I think these changes improve the work and manuscript, whose main contribution is now a new resource comprising a large pool of putative structured RNA regions within coding and non-coding regions adjoined by additional genomic information about them (as opposed to a new approach to finding structured RNAs presented previously). It is evident that much work and computational resources were put into the new set of analyses, and given the computational complexity of such searches, I believe this will provide a useful resource to the RNA biology community.

My only comment is that the new manuscript title seems to imply that the study identified "valid" structured RNAs whereas they are all putative at this point and yet to be validated.

It is also worth noting that I greatly appreciated reviewer 1's thorough review and constructive comments, from which I have learned a lot.

Authors Response

Point-by-point responses to the reviewers' comments:

Reviewer #1: This paper is vastly improved in comparison to the previous version. I commend the authors on their response! I'm fundamentally in support of this paper being ready for publication. However, there's a few claims related to the FDR estimates that I think are not entirely justified. I think these issues can be addressed with some small edits to the text, as the analysis in the paper seems fundamentally reasonable to me for a publication. Additionally, I also have a list of further minor points that I think can also be addressed by simple rewrites (except for a potential bug in the drawings, which however does not seem to affect the underlying analysis).

We genuinely appreciate the reviewer's praise of our work and deeming it fundamentally ready for publication. The substantial improvements would not have been possible without the thorough feedback we received. We also appreciate that the reviewer believes the choices of different tools used is of interest and can provide value to researchers. Again, the reviewer's feedback inspired us to consider proposing models with more covariation and to use tools like R-scape to better assess covariation and SISSIz to place a lower bound on the FDR. The flaws in FDR estimates were overlooked by us, but this is a limitation we have included in the updated manuscript.

As the Discussion essentially states, the results in this paper are a first step towards finding RNAs in these locations. I think that a significant amount of work will be needed to actually find good candidates for experiments, and that the true FDR is likely much higher than what was estimated, as I explain below. However, I think the analysis performed in this paper is well done and is a meaningful step towards this goal. Moreover, in my opinion, the paper makes a number of interesting choices in which tools to use and how to combine results that I think will be of interest to others.

We agree that a significant amount of work is necessary to further filter these candidates for experimental follow up. We are currently working on followup experiments to determine which structured RNAs may be ncRNAs that bind to target genes (and also those likely to be cis-regulatory). We are also further filtering to determine which candidate structured RNAs are encoded in and are being expressed in genetically tractable organisms. This will enable us to do more detailed gain-of-function and loss-of-function experiments to characterize the functions of specific candidate structured RNAs.

ABOUT THE FDR ESTIMATES

The paper now does an overall very reasonable and quite thoughtful job of an analysis of FDR with the SISSIz tool. However, I think it's a bit simplistic to assume that the estimated FDR is close to the true FDR. The main issue is that the shuffled alignments do not take into account sources of false positives due to other biological factors that look somewhat like RNAs. These other factors can bias alignments to look more like RNAs in a way that the SISSIz model doesn't take into account. These factors include palindromic binding sites of dimeric proteins, Rho-independent terminators, repeats & remnants of repeats, etc. It is not, of course, feasible to fully model all such features -- many of them are probably not even yet known to science. So, I think the paper's analysis is fundamentally good. But the FDR estimate is nonetheless not likely to be reliable. A arbitrarily looked at the first few predicted candidates (by ID number), and most seem dubious to me:

- HMP2_1 : this looks like a Rho-independent terminator (so, it is at least an RNA, but there's no evidence of a new RNA type)
- HMP2_2 : The first 3 sequences look like another Rho-independent terminator, and the 4th and 5th sequences do not seem homologous (which negates considerations of covariation, as covariation is meaningless without a homology relationship)
- HMP2_9: the first stem does not exhibit covariation, although it does look somewhat like a terminator. The second stem is highly erratic, is missing in many sequences, and the little covariation there is seems to arise from arbitrary shifts in some very A-U-rich sequences.

- HMP2_10: the covariation in the 3' half of the alignment seems to mostly arise from sequences that don't seem homologous. The stem contained in the 5' half of the alignment has more energetically unfavorable base pairs than examples of covariation, although it could be valid. Anyway, these observations don't seem to suggest an FDR of 2.5-5%. I also see a possible subtle bias in the way the tests with SSISSz were done, although I don't see a feasible strategy to get around this problem. First, I assume CMfinder produced alignments based on entire intergenic regions, but the re-analysis of the SSISSz shuffled alignments used only the part that was incorporated into the alignment. Moreover, the real CMfinder alignments had an opportunity to look at all intergenic regions to potentially find false positives, whereas the FDR estimates only looked at a much smaller set of sequences (those in alignments chosen by CMfinder and the various filters). It's a bit like if we flipped 2 coins 1000 times, and found that both coins were heads in 250 tosses. Then we try to estimate a background probability for the 250 double-heads by re-flipping the coins in the 250 examples, and get 62 re-flips with both heads -- we should have used the original 1000 tosses. I don't think it's feasible to any better than the current paper's analysis, because analyzing all intergenic regions would require somehow randomizing all metagenome sequences while precisely preserving homology relationships (which seems very hard and is why SSISSz requires an alignment). It also wouldn't solve the problem of confounding biological phenomena. Also, I found it great that the paper repeats the full pipeline on the shuffled alignments, so that they are as least as comparable as possible. So, once again, I think the paper did a good job with the analysis, but this might be an additional reason why the FDR estimates could be underestimated. To be clear, I think the paper does as well as possible to control for issues, and I'm not suggesting any kind of mistake. (Also, I should say that the paper I recommended in my previous review has analogous problems that aren't discussed in that paper; looking back, I should have mentioned these problems more clearly in my previous review of the current paper.) Nonetheless, these are the limits of the current state-of-the-art, and I think the paper should be more careful in claiming that the predictions are high quality, or that it's likely that there really are only 2.5-5% false positives. So, after all that, basically I recommend rewording the "high confidence" claims throughout the paper. Something like a "carefully constructed automated analysis", for example. Perhaps the paper could briefly mention potential problems with the FDR estimates (no need to go into as much detail as I have here), and describe them as a rough lower-bound on the true FDR rate.

We thank the reviewer for this detailed explanation of flaws in FDR calculations and appreciate the intuition from inspecting candidate structures. It makes sense that we would be underestimating FDR given this thoughtful assessment. We removed all statements declaring predictions as "high-confidence". We like the phrasing "carefully constructed automated analysis" and have included it in the abstract. When we discuss the FDR in the results, we directly follow it with "These calculations of FDR are likely an underestimation or lower bound of the true FDR."

We also elaborate on this point in the limitations section of our Discussion: "Fourth, the true FDR of these candidate structured RNAs is likely higher than our calculations. Since we predict a diverse set of candidate structured RNAs, we cannot expect shuffling of alignments to adequately control for all biological features. Additionally, we are only shuffling alignments specific to candidate structured RNAs and not shuffling the entire intergenic space. The FDR we estimate in this work is best interpreted as a lower bound of the true FDR." As recommended by the reviewer, we did not go into as much detail as the above review, but we wanted to at least touch on the reasons why the true FDR is likely higher.

Additionally, I find this sentence a problematic because of the difficulties with accurate FDR estimates: "This is a substantial contribution of candidate structured RNA families; for reference, Rfam 14.3 [15] currently contains 3,446 families." Most Rfam families have gone through a much more stringent process than those in the paper under review. (Although, I'll admit that some of the early E. coli RNAs got included in Rfam too easily.) I don't think it's appropriate or meaningful to compare the 3161 predictions to the 3446 families in Rfam.

We removed this sentence from the manuscript. We agree that these candidate structured RNAs should not be directly compared to more rigorously validated Rfam models in this way.

I should also say that, despite my caveats, I do find the FDR estimates to be useful, as they give some indication that the analysis is likely to be good. Even if they might be a kind of lower bound on the true FDR, I think it's an important control to do this kind of analysis -- if the FDR had been estimated at 90%, we'd know there was some kind of serious problem, whereas the low apparent FDR at least says that many of the predictions are likely to be meaningful. For this reason, I think the revised past is a vast improvement over the previous version (which had no controls), and I'm basically in support of publication.

We thank the reviewer again for this suggestion. Even from our perspective, the FDR was extremely helpful to us as we decided on thresholds for various tools. It gave us some baseline to assess how well the workflow and thresholds at various points were performing and gave us confidence in the decisions we were making.

MINOR COMMENTS RELATED TO COMPUTATIONAL LIMITATIONS

I felt there were a few statements that were a little off related to the claim that the major problem with the basic comparative approach is computational limitations.

The paper makes claims in a few places similar to the following: "While powerful, a drawback of this comparative genomics approach is that in any given experiment, only a select set of intergenic regions are typically considered; this is done to accommodate computational limitations. For example, comparative genomics approaches often subset intergenic regions to bias towards known classes of structured RNAs, like ribozymes or specific types of riboswitches, which tend to be found in specific genomic contexts [2]." The relevant part of ref [2] reads "However, the complete set of available genomic and metagenomic IGRs totals 37 billion base pairs, which presents two difficulties. First, the computational time and memory required for analysis expands with many IGRs. More crucially, the false positive rate when IGRs are compared increases with their total number, making it difficult to extract rare RNAs." The paper under review seems to ignore the sentence beginning "more crucially". I would guess that HS-BLASTN has a high specificity and relatively low sensitivity (in order to be fast), and this reduces the false positives in matches. Unfortunately, this presumably also reduces sensitivity. I think the paper under review should also briefly mention the challenge of false positives or false negatives in analyzing huge datasets, which might compromise the ability to actually find RNAs in huge datasets.

Thank you for pointing this out. We added a discussion of the false positives/negatives to the introduction in addition to the FDR limitations added to the discussion.

Regarding the reviewer's comment on HS-BLASTN: We performed BLASTN on a small subset of the intergenic regions (to estimate how long it would actually take to run) and we saw identical results between BLASTN and HS-BLASTN. The HS-BLASTN aims to "accelerate MEGABLAST and produce identical results" (Chen et al, 2015). This seems to be true (identical results between the two) from the overlap we have observed; however, there have been others who have noticed reduced sensitivity (<https://github.com/chenying2016/queries/issues/10>). Given this, we are not entirely sure if HS-BLASTN indeed results in reduced sensitivity (in theory it should not) but this could be possible if we did a more thorough comparison of all intergenic regions with BLASTN.

(2) The paper under review more or less solves this computational load problem, but doesn't explicitly say this. The first paragraph of Discussion gets at this, even citing ref [2] again, but doesn't discuss what the innovation was, or explicitly say that there was no need to limit intergenic region (except that only human microbiomes were used). My sense is that the major innovation is the use of HS-BLASTN (combined with a lot of CPU hours). Some more direct statement on this in the Discussion would be great.

We thank the reviewer for this suggestion. We add an additional paragraph to the discussion elaborating on HS-BLASTN.

"A key tool that enabled us to predict candidate structured RNAs at a large-scale was HS-BLASTN. A key distinction between HS-BLASTN and BLASTn is that HS-BLASTN utilizes a new database-derived lookup table. Unlike BLASTn, it loads the resulting index into memory and thus requires significantly more RAM. To perform all vs. all HS-BLASTN in this workflow required roughly 35,000 CPU-hrs at 90 GB of RAM. We estimated that BLASTn would have required an order of magnitude less RAM but also over an order of magnitude more CPU-hrs. Though this work was still computationally intensive, it would have otherwise been too intensive to realistically perform without HS-BLASTN."

MISCELLANEOUS MINOR COMMENTS

I'm not 100% clear on how the RNA-seq analysis for RNA candidates was conducted. I think what happens is that the locations of instances of each alignment are compared to regions that are expressed in RNA-seq data. Is this true? What happens if a sequence belonging to the alignment is only partially expressed in the RNA-seq data (e.g., the 5' half overlaps RNA-seq reads, but not the 3' half)? If a large region is expressed according to RNA-seq, and a prediction covers a part of this, I assume that counts as expressed, correct? Basically, I assume that if one nucleotide of an RNA candidate overlaps one nucleotide of an RNA-seq read, then it's considered to be expressed, right? I'm not suggesting any potential problem here, just that it would be nice to be more explicit about this in the paper. (Could also be supplementary text.)

Thank you for this comment. We should have been more clear here. Your intuition is correct. We added this to the methods: "If an RNA-Seq read mapped to any position that overlapped with a predicted structured RNA (even at a single position), the read was counted." Thus, with this simple approach, we did not check for uniform coverage across the structure. This was only a very simple analysis counting how many reads overlapped any position along the structure and then normalizing the count. Many of the highly expressed RNAs did have signal very specific to the boundaries of the structure (some of these are shown in the figures). Though you can't assume this to be true based on the RPKM counts we provide.

It might be appropriate to cite GraphClust (<https://www.ncbi.nlm.nih.gov/pmc/articles/PMC3371856/>) and/or GraphClust2 (<https://www.ncbi.nlm.nih.gov/pmc/articles/PMC6897289/>) as another approach to clustering large datasets, perhaps with a brief sentence. This program uses locality-sensitive hashing, and the amount of time it takes is theoretically linear in the number of input sequences, as opposed to quadratic with BLAST. To my knowledge, GraphClust has not actually been applied to truly huge datasets, but at least in principle it could be.

This is a great suggestion. We now propose this as a future direction in our discussion as a possible way to predict additional structures.

The paper analyzes genes within 5 Kb of RNA instances, which as I would understand it, includes genes upstream and downstream. What's the biological motivation for this? I think it's rare that an RNA's function has a strong association with genes that are nearby in either the 5' or the 3' direction. Also, the statistic fails to take into account the fact that some genes are common, e.g. HTH-XRE. The paper could leave the existing analysis with some kind of rationale, and just point out that it doesn't account for gene frequency. If the paper has a particular goal in citing this statistic, it might be good to normalize the gene frequency near to the RNA by dividing it by the total number of gene occurrences in the dataset. I'd recommend simply removing the analysis entirely.

Thank you for this comment. There are a few reasons why we decided to include the 5 kb analysis. First, we found it very helpful as we tried to make sense of which candidate structured RNAs were interesting. For example, the CRISPR repeats in Figure 5 were first identified by realizing that they most strongly associated with Cas genes within 5 kb of the repeats. Additionally, there were several structures throughout the manuscript that we were able to identify as phage-associated because they were surrounded by phage protein domains (we referred to several of these throughout the manuscript). Taxonomic classifications struggle to identify phage and often report the taxonomic classification as the bacteria the phage is often found in (so they are classified to the wrong taxonomic domain). More broadly, cis-regulatory structured RNAs can be found upstream and downstream of genes and we do not have a way to estimate the UTR lengths. To identify possible cis-regulatory structured RNAs, we looked 25 bp upstream of genes, which is arbitrary. Because there are no good thresholds, we liked the idea of casting a wide 5kb net in both directions. Sometimes we get strong and meaningful associations of structures to genes this way (like Cas protein domains), though we do agree that genomic neighborhoods oftentimes will not tell us much about the function of a structured RNA.

HTH-XRE RNA motifs are a good example of this problem as you mentioned. There is a category of structured RNA motifs that occur directly upstream of HTH-XRE and even we are predicting 15 additional candidates within 25 bp upstream of HTH-XRE (https://en.wikipedia.org/wiki/HTH-XRE_RNA_motif). However, it is unclear if these structures are actually cis-regulatory, especially considering the 5 kb analysis where 100 candidates were within 5 kb of this domain (it is just a common domain). Perhaps these structures just happen to frequently be upstream of HTH-XRE by random chance. The 5 kb analysis also puts the 25 bp analysis into perspective in this way. There is no good way to control for protein domain frequencies except by running RPS-blast on every gene in HMP2, which would be intensive and also as you mention probably not worth it since that analysis adds limited value.

In the results, we now added a bit more about the usefulness of the 5kb analysis; for the reason of better taxonomy alone, it is probably worth keeping the 5 kb analysis as it helps suggest which structures might be mobile and on phage. In the limitations, we now addressed the issue of protein domain frequency and how genomic neighborhood is not that useful if the structure does not have cis-regulatory functions.

I don't see a description of how CMfinder was used for the predictions. As I understand it, CMfinder normally produces multiple predictions. What command line was used / how was it processed?

This is a good point that we have now addressed in the methods. In cases in which CMfinder produced multiple predictions and more than one of those predictions met the thresholds of p-score and covariation, we chose the alignment with the highest p-score. We did not want to propose multiple models for the same region as this seemed redundant and inefficient. Another choice we could have made is to choose based on the number of covarying bases from R-scape; however, this very often did not result in a clear cut winner and eventually lead us to breaking the tie with p-score anyway. To our knowledge, there is no standard way to approach this; however, p-score seemed like an appropriate decision.

It'd be nice if the paper would report the total number of basepairs in the set of intergenic regions that were analyzed.

Thank you for the feedback. There were a little over 22 million base pairs considered in the intergenic regions. We added this to the manuscript.

The response to review states that it took 35,000 CPU hours at 90 GB RAM to do the HS-BLASTN comparisons. I think this information would be interesting for the paper, and perhaps the other information in the paragraph of the authors' response (why NCBI BLAST wouldn't be feasible, and something like the statement "The trade off of this program is fewer CPU-hours necessary at a cost of higher RAM demand.") I actually read the paper before looking at the responses, and wanted to know more about the resources required by HS-BLASTN, and then saw that was discussed in the response to my review.

This is a great suggestion that should be included in the paper. We included a paragraph in the discussion as well as some additions to the introduction to elaborate on this.

What was the rationale for filtering HS-BLASTN matches as described (≥ 30 bp, $< 100\%$ identity, < 100 bit score)? I don't see any need to validate these choices, but a statement about why they were chosen would be nice, even if it was just to say, for example, that it was based on the authors' intuition. In filtering HS-BLASTN results, was there a minimum score, or some kind of threshold to remove meaningless matches? What was the relevant default setting in HS-BLASTN (in terms of a score or E-value threshold)?

We really appreciate this catch as this is a critical threshold. We set the E-value cutoff to 0.05. Most of our inspiration for these thresholds came from this previous Genome Biology paper (<https://genomebiology.biomedcentral.com/articles/10.1186/gb-2010-11-3-r31>). We now made this more clear in the manuscript. They also removed sequences that were self-matching and with BLAST scores > 100. To reduce false positives, they only considered intergenic regions of at least 30 nucleotides and we further extended this to requiring conserved regions of at least 30 nucleotides. This was mostly based on the existing intuition and also it indeed seemed to help our FDR estimates. Notably in their paper, they manually defined the cutoffs of blast scores, choosing $S = 35, 40, \dots, 85$...essentially focusing on hits within a score (S) range of X to 100, and X was manually changed as needed. Since it was difficult to scale this manual decision, we went with a safe bet of E-value less than 0.05 (which we considered a defensible number for similarity) and still kept the $S=100$ upper bound. Overall, most of these decisions were a result of using the intuitions of this previous paper and intuitions we were gaining by the trial and error of trying different thresholds and estimating FDR.

I'm unclear on how the fasta files were split for HS-BLASTN, with this sentence from the manuscript: "we split the intergenic regions into 100 roughly equal smaller fasta files and ran HS-BLASTN [8] on each file before combining to accommodate RAM requirements." First, I think "to accommodate RAM requirements" belongs elsewhere in the sentence (or perhaps this could be broken into more than one sentence). More importantly, I'm not sure I get the idea. Was HS-BLASTN initially used on one fasta file to remove highly similar sequences within that file, so that the total number of sequences is fewer? If so, that would certainly reduce the number of comparisons, but how was it defined whether sequences were too similar (and could be removed)? The stated filters described earlier in the paper (≥ 30 bp, $< 100\%$ identity, < 100 bit score) filter the matches, not sequences. If sequences were not removed from the individual fasta files, then I don't see how processing the individual

fasta files would help to reduce computer time. I think there's one or two pieces of information missing here.

Thank you for this comment as this could be written better. We have reworded the methods section of the manuscript. One of the disadvantages of HS-BLASTN (from their manuscript) is that it "loads the FMD-index into RAM before running the search procedure." The size of the FMD-index depends on the size of the genomic database. If we were to try to use the entire intergenic space all at once as the database, it would take an unrealistic amount of RAM to search against. If we build a database with 1/100 of the intergenic space, it required under 90 GB of RAM, which was realistic for us. We first broke up the database in an unbiased way into 100 relatively equally sized fasta files and made 100 databases. We then queried all intergenic sequences against all 100 databases. Then, we combined the blast results. With standard BLASTN, we could have used the entire intergenic space as a database (because we don't need that much RAM even for all intergenic sequences), but it would have taken significantly more CPU-hrs to run. The tradeoff to making this run in a reasonable time frame was to use significantly more RAM and to break the intergenic space into smaller databases (1/100 size) that could be independently loaded into RAM. Of course, if we had easy access to huge amounts of RAM it may have been possible to make one giant database and load it into RAM for HS-BLASTN.

"Rfam 14.31" is mentioned in one case. I assume that version 14.3 is meant.

Thank you for identifying this. The typo was corrected to 14.3.

What does the yellow shading in Fig 2D mean?

Thanks for pointing this out. The yellow shading represents the mRNA component of the tmRNA. From the diagram, you can see the start and stop site, and we figured we would just highlight it. We added a statement on this in the legend.

How was R2R used in this paper? I am surprised by the extremely small number of covarying base pairs in most of the figures within the main text. Moreover, since R-scape was used in the filter, it's surprising if there are really no covarying base pairs at all in some of the candidate RNAs (HMP2_3156 in Fig 3D, HMP2_4078 in Fig 4A, HMP2_1065 in Fig 4C). Looking at the raw alignments, it seems like there should have been covariation according to R2R or R-scape. So it seems like something weird is going on with converting the alignments to drawings.

Thanks for pointing this out. We ran R-scape with default settings. R-scape also outputs results from R2R, which is how the visuals were created. We now specify this in the methods. The three structures you indicate do have 1 covarying set of bases highlighted in green in those structures. If you look at raw alignments, it indeed seems as though our R2R diagrams are substantially underestimating the amount of covariation in structures. Only those deemed significant by R-scape ($E < 0.05$) are being highlighted green and signs of covariation that are not predicted to be as significant are not being shown. In terms of covariation, the drawings are just very conservative depictions of the alignments. We could run R2R with different parameters than R-scape to produce models with visually more covariation, but we figured it was better to keep the covariation predicted by R-scape and the R2R diagram consistent.

Fig 2: some nucleotides are difficult to read because they overlap.

Thank you for pointing this out. We adjusted the images and also manually corrected some of the overlapping bases.

Reviewer #2: The authors made substantial changes to their analysis and consequently to the focus of their work and manuscript. Specifically, they addressed the comments and flaws previously pointed out, which led them to design a more robust and reliable pipeline for structured RNA detection via standard comparative genomic analysis methods. I think these changes improve the work and manuscript, whose main contribution is now a new resource comprising a large pool of putative structured RNA regions within coding and non-coding regions adjoined by additional genomic information about them (as opposed to a new approach to finding structured RNAs presented previously). It is evident that much work and computational resources were put into the new set of analyses, and given the computational complexity of such searches, I believe this will provide a useful resource to the RNA biology community.

We greatly appreciate the reviewers belief that we adequately addressed previous flaws and have made substantial progress to create a useful resource to the RNA biology community. We thank you again for your concerns on the stringency of our models in the last submission as this has allowed us to dramatically improve our work.

My only comment is that the new manuscript title seems to imply that the study identified “valid” structured RNAs whereas they are all putative at this point and yet to be validated.

We appreciate this comment. This was also one of the major concerns of reviewer 1 and rightfully so. The ways we have addressed this are:

Using “candidate” in front of structured RNAs more diligently. The title of the manuscript especially was too strongly worded and now makes it clear that these are candidates.

We better address limitations in FDR estimations.

We avoid suggesting that these predictions are “high-confidence”.

It is also worth noting that I greatly appreciated reviewer 1’s thorough review and constructive comments, from which I have learned a lot.